# Centrosome amplification primes ovarian cancer cells for apoptosis and potentiates the response to chemotherapy

**Frances Edwards**[1]\*, **Giulia Fantozzi**[1], **Anthony Y. Simon**[1], **Jean-Philippe Morretton**[1], **Aurelie Herbette**[2], **Andrea E. Tijhuis**[3], **Rene Wardenaar**[3], **Stacy Foulane**[1], **Simon Gemble**[1], **Diana C.J. Spierings**[3], **Floris Foijer**[3], **Odette Mariani**[4], **Anne Vincent-Salomon**[4], **Sergio Roman-Roman**[2], **Xavier Sastre-Garau**[4], **Oumou Goundiam**[2], **Renata Basto**[1]\*

**1** Biology of centrosomes and genetic instability, Institut Curie, PSL Research University, CNRS UMR 144, Paris, France, **2** Department of Translational Research, Institut Curie, PSL University, Paris, France, **3** European Research Institute for the Biology of Ageing, University of Groningen, University Medical Center Groningen, Groningen, the Netherlands, **4** Department of pathology, Institut Curie, Paris, France

\* frances.edwards@curie.fr (FE); renata.basto@curie.fr (RB)

**Data Availability Statement:** All relevant data are within the paper and its Supporting Information file. Data underlying this study are from the Biological Resource Center (BRC) of Institut Curie

## Abstract

Centrosome amplification is a feature of cancer cells associated with chromosome instability and invasiveness. Enhancing chromosome instability and subsequent cancer cell death via centrosome unclustering and multipolar divisions is an aimed-for therapeutic approach. Here, we show that centrosome amplification potentiates responses to conventional chemotherapy in addition to its effect on multipolar divisions and chromosome instability. We perform single-cell live imaging of chemotherapy responses in epithelial ovarian cancer cell lines and observe increased cell death when centrosome amplification is induced. By correlating cell fate with mitotic behaviors, we show that enhanced cell death can occur independently of chromosome instability. We identify that cells with centrosome amplification are primed for apoptosis. We show they are dependent on the apoptotic inhibitor BCL-XL and that this is not a consequence of mitotic stresses associated with centrosome amplification. Given the multiple mechanisms that promote chemotherapy responses in cells with centrosome amplification, we assess such a relationship in an epithelial ovarian cancer patient cohort. We show that high centrosome numbers associate with improved treatment responses and longer overall survival. Our work identifies apoptotic priming as a clinically relevant consequence of centrosome amplification, expanding our understanding of this pleiotropic cancer cell feature.

## Introduction

Centrosomes are the major microtubule organizing centers (MTOCs) in proliferating animal cells, whose structure and number are tightly regulated during the cell cycle [1]. The centrosome is duplicated during S-phase in a PLK4 kinase-dependent manner and the 2 centrosomes contribute to the timely and functional assembly of a bipolar spindle during mitosis. In cancer

(certification number: 2009/33837.4; AFNOR NF S 96 90). The access to the tumor cohort is restricted and subjected to the ethical evaluation by the Institut Curie. To inquiry about this cohort please consult https://institut-curie.org/, or contact the head of the pathology department at Institut Curie: Dr Anne Salomon: anne.salomon@curie.fr.

**Funding:** Labex CelTisPhyBio (https://labex-cellnscale.institut-curie.org/) provided the post-doctoral fellowship and Agence pour la Recherche contre le Cancer (www.arcfoundation.org) provided the postdoctoral fellowship PDF20190508563 to FE. La Ligue contre le cancer (www.ligue-cancer.net) PhD fellowship 17562 to GF. This work was supported by an InCA-Bio (www.e-cancer.fr) (2015-PLBIO15-237 grant, a WWC (www.worldwidecancerresearch.org) grant, (21-0042), an InCA (www.e-cancer.fr) (2021-1-PREV-Bio grant), the Institut Curie and the CNRS. The Basto lab is a member of the CelTisphybio labex (https://labex-cellnscale.institut-curie.org/). The funders did not have any role in study design, data collection and analysis, decision to publish, or preparation of the manuscript.

**Competing interests:** RB is a member of the PLOS Biology Editorial Board.

**Abbreviations:** BRC, Biological Resource Center; CART, Classification and Regression Trees; CNR, centrosome to nuclei ratio; dox, doxycycline; HGSOC, high-grade serous ovarian cancer; HPF, high-power field; MOMP, mitochondria outer membrane permeabilization; MTOC, microtubule organizing center.

cell lines, centrosome structural and numerical defects are common [2] and in particular centrosome amplification—more than 2 centrosomes per cell—has also been observed in situ in tumor samples [3–5]. Cells with centrosome amplification perform bipolar mitosis via centrosome clustering mechanisms during spindle assembly [6–10]. Centrosome amplification is nevertheless associated with chromosome instability [11,12] and increased invasive behaviors [13–15]. As first postulated by T. Boveri [16], centrosome amplification can drive tumorigenesis in vivo [7,17–19]. Defects in centrosome clustering capacity are associated with lethal multipolar divisions [8,11,20], motivating the search for inhibitors that limit centrosome clustering and eliminate cells with centrosome amplification [20–25].

Carboplatin and Paclitaxel are cytotoxic agents used in treatment of various solid cancers, and their combination is the standard of care for treatment of advanced stage epithelial ovarian cancer patients [26]. Carboplatin induces DNA damage and Paclitaxel stabilizes microtubules leading to cell death via mitotic catastrophe which is defined as death during or following abnormal mitosis [27,28]. Despite the central role of centrosomes in spindle assembly, how centrosome amplification influences the response to combined Carboplatin and Paclitaxel remains unexplored. Paclitaxel has been shown to induce multipolar divisions [29] and this can be increased by centrosome amplification [30]. The impact of centrosome amplification on the response to DNA damaging agents has however not been explored despite centrosomes regulating multiple signaling pathways that could influence chemotherapy responses [31–35]. Multiple consequences of centrosome amplification could therefore synergize with combined Carboplatin and Paclitaxel to induce efficient cancer cell elimination.

Here, we chose to study how centrosome amplification influences the response to combined Carboplatin and Paclitaxel in the context of epithelial ovarian cancer, a disease with poor clinical outcome related to late diagnosis, and frequent relapse [36]. Centrosome amplification is observed in ovarian cancer cell lines, and we recently also characterized its occurrence in situ in patient samples [5]. We use an inducible PLK4 overexpression system in ovarian cancer cell lines to induce centrosome amplification in isogenic backgrounds. We perform single-cell live-imaging of cells to assess the correlations between mitotic behaviors and cell fate during chemotherapy. We show that centrosome amplification potentiates the response to combined Carboplatin and Paclitaxel via multiple mechanisms. Beyond multipolar divisions associated with Paclitaxel exposure, we found that centrosome amplification also enhances cell death independently of mitotic behaviors. We show that centrosome amplification, although well tolerated by ovarian cancer cells, leads to mitochondria outer membrane permeabilization (MOMP) priming. We assess the level of centrosome amplification in a previously characterized ovarian cancer patient cohort and observe an association between high centrosome numbers and the patient time to relapse as well as their overall survival. Together, our work shows for the first time that centrosome amplification can synergize with combined chemotherapy, advancing our understanding of its consequences in cancer.

## Results

### Centrosome amplification enhances cell death in response to combined Paclitaxel and Carboplatin

To study the influence of centrosome amplification on the response to Carboplatin and Paclitaxel, we used an inducible PLK4 overexpression system (PLK4OE) in the epithelial ovarian cancer cell line OVCAR8. Exposing cells to doxycycline (dox) for 72 h at 1 μg/ml induced centrosome amplification (more than 2 centrosomes per cell) in around 80% of cells, compared to 4% in control OVCAR8 (DMSO treated, PLK4Ctl) (S1A and S1B Fig). We used MTT viability assays to determine Paclitaxel and Carboplatin IC50s over the 72 h following PLK4OE

induction (S1C Fig). For both drugs, the IC50 was lower for PLK4OE (67 μm and 3,4 nM, respectively) compared to PLK4Ctl (136 μm and 5,1 nM, respectively), already suggesting that centrosome amplification sensitizes cells to these chemotherapeutic agents. We also determined working combination concentrations (100 μm Carboplatin + 3,3 nM Paclitaxel) that induce 60% growth inhibition in PLK4Ctl and 89% in PLK4OE (S1D Fig).

We next used live-imaging to investigate how centrosome amplification influences the response to chemotherapy and in particular if it increases catastrophic mitosis. We used H2B-RFP expressing OVCAR8 allowing us to observe chromosome behaviors during mitosis and chromatin compaction that occurs when cells die (Fig 1A and S1 Movie). We imaged cells during the 72 h of exposure to 100 μm Carboplatin + 3,3 nM Paclitaxel and performed analysis of complete cell lineages, counting the number of cells produced per lineage (proliferation) and the fate of these cells (viability). In untreated OVCAR8 cells, PLK4OE reduced proliferation compared to PLK4Ctl, but independently of an increase in cell death (Fig 1B and S2 Movie). Combined chemotherapy reduced proliferation of both PLK4Ctl and PLK4OE cells and this reduction was associated with an increase in cell death (Fig 1B and S3 Movie) as well as cell cycle lengthening (S1E Fig). In agreement with the MTT dose-response assays, combined chemotherapy induced a higher proportion of cell death in PLK4OE with 75% of cells dying, compared to PLK4Ctl where 33% died (Fig 1B, 1E, and 1F). By examining cell fate in consecutive generations, we observed that cell death was mainly occurring in generations 2 and 3 suggesting that passage through mitosis or extended exposure time to chemotherapy is detrimental for the progeny (Fig 1C).

We therefore characterized the mitotic behaviors in the first generation focusing on chromosome mis-segregation (Fig 1A and S4 Movie). In untreated PLK4Ctl OVCAR8, we observed a significant proportion of chromosome instability with around 20% of mitosis occurring with either chromosome mis-alignment, one lagging chromosome, or one chromatin bridge (Figs 1A, black arrows, S1F and S1G). These behaviors, which are sometimes difficult to discriminate at the low spatiotemporal resolution used in our long-term live-imaging approach, were pooled together and considered as Slight mis-segregation events (Fig 1A). These events were more frequent in PLK4OE (36%), in agreement with centrosome amplification inducing merotelic attachments [11]. However, the proportion of Multipolar divisions was negligible despite the presence of supernumerary centrosomes (S1F and S1G Fig). Importantly, combined chemotherapy induced an increase in 2 behaviors associated with strong chromosome mis-segregation: Multipolar divisions and bipolar divisions associated with multiple chromosome mis-segregation events (combinations of bridges, lagging and misaligned chromosomes—termed here High mis-segregation) (Fig 1A 1D vertical axis and 1E and 1F). Events of complete division failure—either via cytokinesis failure, mitotic slippage, or death in mitosis—were observed but remained infrequent upon combined chemotherapy (Fig 1D, vertical axis). PLK4OE cells exposed to combined chemotherapy present close to 47% of multipolar divisions compared to 6% observed in PLK4Ctl (Fig 1D vertical axis and Fig 1E and 1F). We next focused on the fate of the progeny produced by the different cell division categories and observed that within cells completing cell division, higher levels of chromosome mis-segregation were associated with increased cell death in the progeny (Fig 1D, horizontal axis). In particular, multipolar divisions were associated with at least 50% cell death in the progeny, and the increase of these multipolar divisions in PLK4OE cells exposed to combined chemotherapy therefore contributes to the decreased viability observed in this condition. Paclitaxel has been suggested to induce multipolar spindles in cells that present centrosome amplification [30], and exposing PLK4OE cells to Paclitaxel alone also induces multipolar divisions (S1H Fig), suggesting the increased multipolarity we observe in presence of the combined chemotherapy is caused by Paclitaxel. Indeed, staining mitotic spindles for α-Tubulin and centrosome

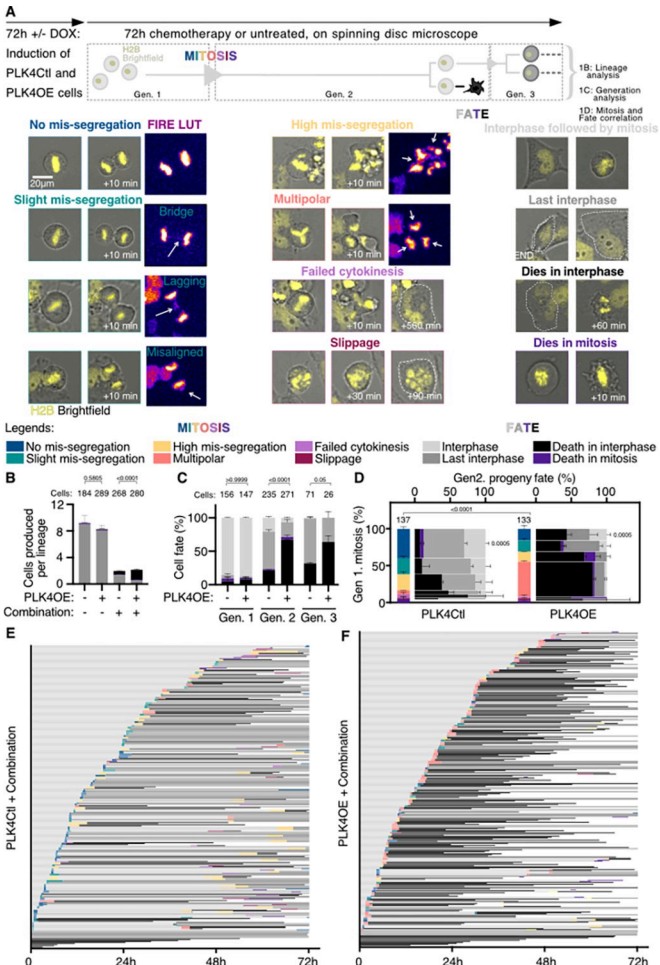

**Fig 1. Centrosome amplification enhances cell death in response to combined chemotherapy. (A)** Single-cell live-imaging workflow. OVCAR8 cells expressing H2B-RFP and inducible for PLK4 overexpression are exposed to DMSO or 1 μg/ml Doxycycline (DOX) for 72 h to induce centrosome amplification. PLK4Ctl and PLK4OE cells are then filmed during 72 h of chemotherapy and lineages are tracked over multiple generations. Representative images of mitotic behaviors and cell fates are shown with the color-coded legends used in the subsequent panels. White arrows point to defects. Lineage analysis consists in counting for each starting cell, the number of cells adopting the different fates (see panel B). Generation analysis consists in determining the percentage of a generation that will adopt the different fates (see panel C). Mitosis and fate correlation consist in determining the percentage of cells adopting the different fates, depending on the behavior of the mother cell during mitosis (see panel D). **(B)** Bar graphs showing the averages and SEM of the number of cells per lineage adopting the indicated fates (legends in panel A). A minimum of 20 lineages were analyzed from 2 independent experiments, statistical tests: Fisher's exact test on the number of cell death events (pooling death in interphase and in mitosis). Numbers on the top of each graph represent the number of cells analyzed per condition. **(C)** Bar graphs showing the average and SEM of the percentages of cells undergoing indicated fates (legends in panel A). Two independent experiments, statistical tests: Fisher's exact test on the number of cell death events (pooling death in interphase and death in mitosis). Numbers on the top of each graph represent the number of cells analyzed per condition. **(D)** Vertical axis: Bar graphs showing the average and SEM of the percentages of mitotic phenotypes (legends in panel A); 137 and 133 cell divisions were analyzed from 2 independent experiments, statistical test: Fisher's exact test on the number of multipolar divisions. Horizontal axis: Bar graphs showing the average and SEM of the percentages of cells undergoing indicated fates (legends in panel A) according to the mitotic behavior of mother cells, with bar width depending on the proportion of cells displaying a given mitotic phenotype. Two independent experiments, statistical test: Fisher's exact test on the number of No mis-segregation progeny (progeny of blue mitosis) dying in mitosis and interphase, *p*-value = 0,0005. **(E, F)** Single-cell profiles of PLK4Ctl (E) and PLK4OE (F) undergoing Carboplatin + Paclitaxel exposure. Each row corresponds to 1 cell (legends in panel A). Data for Fig 1 can be found in S1 Data.

markers revealed that while bipolar and pseudo-bipolar spindles are the most frequent within untreated PLK4OE prometaphases and metaphases, Paclitaxel treatment induces the formation of more multipolar spindles as well as abnormal looking spindles that we named undefined spindles (S1I and S1J Fig).

However, and surprisingly, we also observed that independently of the type of mitosis induced by combined chemotherapy, the proportion of cell death in the progeny was higher in PLK4OE compared to PLK4Ctl (Fig 1D, horizontal axis). This was in particular the case for the progeny of cells that do not show any chromosome mis-segregation, where 40% cell death is observed in PLK4OE compared to only 12% cell death in PLK4Ctl (Fig 1D, horizontal axis). These results suggest that centrosome amplification increases cell death in response to combined chemotherapy independently of multipolarity and chromosome segregation errors.

## Centrosome amplification enhances cell death in response to Carboplatin independently of catastrophic mitosis

We were next interested in understanding why cell death is enhanced in PLK4OE compared to PLK4Ctl when similar mitotic behaviors are observed. We noticed that the IC50 of Carboplatin is lower for PLK4OE compared to PLK4Ctl (S1C Fig, right). We therefore hypothesized that PLK4OE cells may respond differently to DNA damage induced by Carboplatin. We performed live-imaging to better understand how OVCAR8 cells respond to 136 μm Carboplatin, and how centrosome amplification modifies this response (S5 Movie). First focusing on the lineage analysis, we observed that Carboplatin-treated cells have a reduction in proliferation compared to untreated cells and an increase in cell death (Fig 2A–2D compare with S2F Fig for controls and S1E Fig). In agreement with PLK4OE cells being more sensitive to Carboplatin, fewer viable cells were produced in Carboplatin exposed PLK4OE compared to PLK4Ctl, and this was associated with an increase in cell death with 47% in PLK4OE and 30% in PLK4Ctl (Fig 2A–2D). Similar to the combined chemotherapy, most cell death occurred in generation 2 (Fig 2E) and we therefore focused on chromosome mis-segregation during the first mitosis. In both PLK4Ctl and PLK4OE treated with Carboplatin, the main phenotype was an increase in High mis-segregation divisions mainly occurring after 24 h of Carboplatin exposure (Fig 2A, 2B, 2F and 2G compare with S2I Fig for untreated controls). Unlike in the response to combined chemotherapy, Multipolar divisions remained negligible (Fig 2A, 2B and 2F). We further characterized mitosis with High mis-segregation by staining cells fixed at 48 h of Carboplatin treatment, observing α-Tubulin as well as centrosomal, centriolar, and centromeric markers. In agreement with our live-imaging experiment which characterized these divisions as being bipolar we observed mainly bipolar spindles based on the α-Tubulin staining, both in PLK4OE and in PLK4Ctl (S2A and S2B Fig). These spindles were however frequently characterized by the presence of additional MTOCs even in PLK4Ctl cells that importantly do not seem to contribute to forming functional spindle poles. In these cells, these additional MTOCs varied based on the presence of centrosomal and centriolar proteins (S2C Fig) and suggested Carboplatin may be inducing centriole over-duplication as well as premature centriole disengagement as previously described for other sources of replication stress [37]. Despite bipolar spindle organization, these mitosis were characterized by extensive chromosome mis-segregation and in both PLK4OE and PLK4Ctl, a majority of these anaphases are characterized by the presence of acentric chromosomes (S2D and S2E Fig), and most likely generated aneuploidy at lethal levels.

At the low spatial and temporal resolution used to correlate mitotic fates and cell death, the proportions of the different types of mitosis were similar in Carboplatin treated PLK4OE and PLK4Ctl (Fig 2A, >2B and 2F, vertical axis). The increased cell death observed in PLK4OE

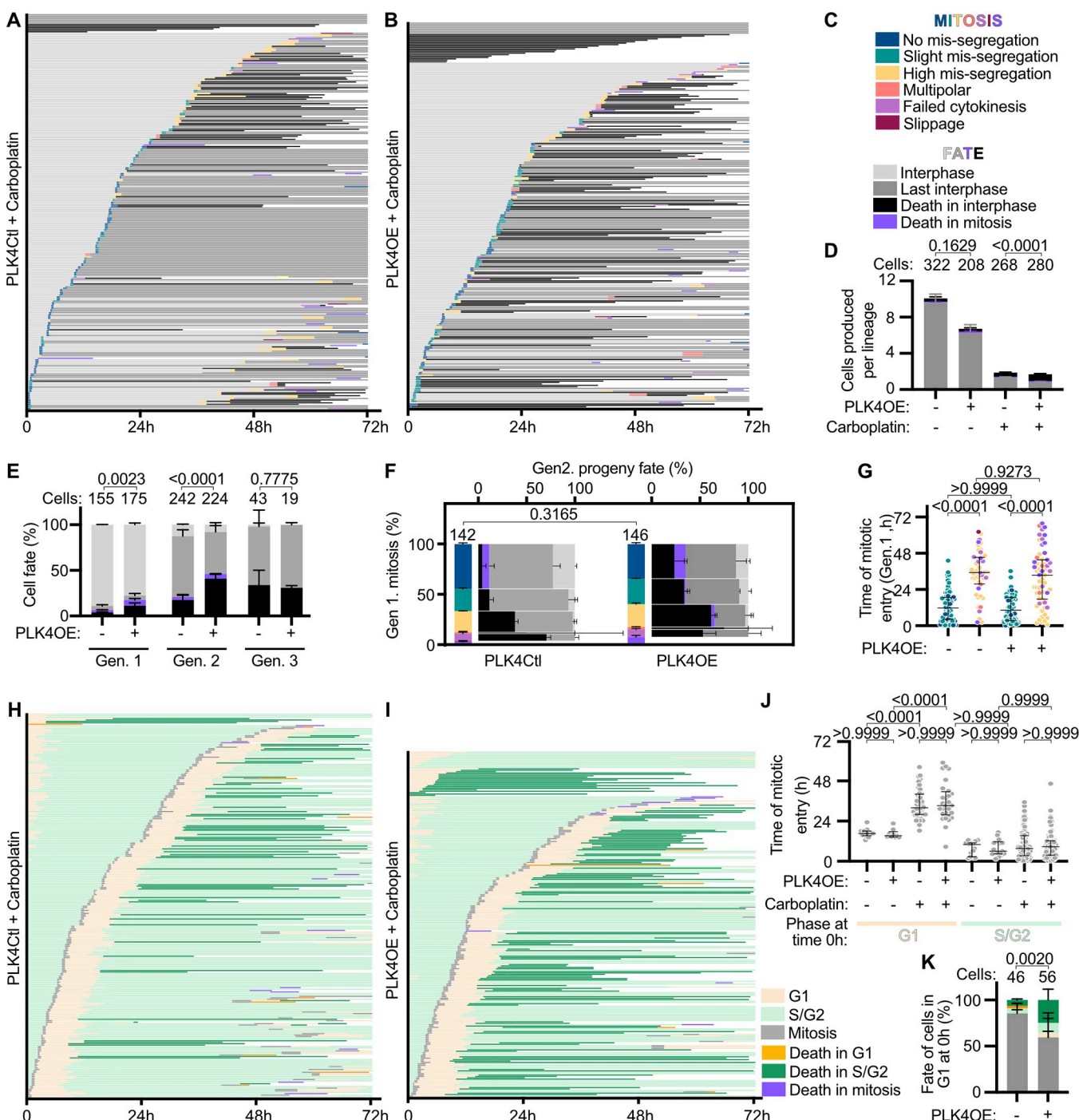

**Fig 2. Centrosome amplification enhances cell death in response to Carboplatin independently of chromosome mis-segregation. (A, B)** Single-cell profiles of PLK4Ctl (A) and PLK4OE (B) undergoing Carboplatin exposure. For comparison with untreated controls—S2F Fig. Color coding of mitosis and fates legends in panel C. **(C)** Legends for panels A, B, D, E, F, and G, as defined in Fig 1A. **(D)** Bar graphs showing the averages and SEM of the number of cells produced per lineage, adopting the indicated fates (legends in panel 1A). A minimum of 31 lineages were analyzed from 2 independent experiments, statistical tests: Fisher's exact test on the number of cell death events (pooling death in interphase and in mitosis). Numbers on the top of each graph represent the number of cells analyzed per condition. **(E)** Bar graphs showing the average and SEM of the percentages of cells undergoing indicated fates (legends in Fig 1A and 1C). Two independent experiments, statistical tests: Fisher's exact test on the number of cell death events (pooling death in interphase and death in mitosis). Numbers on the top of each graph represent the number of cells analyzed per condition. **(F)** Vertical axis: Bar graphs showing the average and SEM of the percentages of mitotic phenotypes (legends in Fig 1A and 1C); 142 and 146 cell divisions analyzed from 2 independent experiments, statistical test: Chi-square test. Horizontal axis: Bar graphs showing the averages and SEM of the percentages of cells undergoing indicated fates (legends in Fig 1A and 1C) according to the mitotic behavior of the mother cell, with bar width depending on the proportion of cells according to their mitotic behavior. Two independent

experiments, statistical test: Fisher's exact test on the number of No mis-segregation progeny (progeny of blue mitosis) dying in mitosis and interphase, *p*-value = 0,0003. **(G)** Scatter dot plot graphs showing time of mitotic entry, with median and interquartile range. Cells were classified depending on mitotic phenotypes with color-code defined in panel C. Two independent experiments with a minimum of 48 mitosis analyzed per category. Statistical tests: Kruskal–Wallis with Dunn's multiple comparisons tests. **(H, I)** Single-cell profiles of FUCCI PLK4Ctl (H) and PLK4OE (I) cells undergoing Carboplatin exposure. Times in G1, S/G2 and mitosis, as well as death in each of these phases are color-coded as indicated. **(J)** Scatter dot plots of the time of mitotic entry depending on cell cycle phase at movie start, with median and interquartile range. Two independent experiments with a minimum of 10 times analyzed per category. Statistical tests: Kruskal–Wallis with Dunn's multiple comparisons tests. **(K)** Bar graphs showing the average and SEM of the percentage of cells adopting the indicated fates (legends in panel I). Two independent experiments, statistical test: Fischer's exact test on the number of cells dying (irrespective of the cell cycle phase). Numbers on the top of each graph represent the number of cells analyzed per condition. Data for Fig 2 can be found in S2 Data.

compared to PLK4Ctl again seems to stem from a difference in the association between cell death and mitotic phenotypes (Fig 2F, horizontal axis). Indeed, in PLK4Ctl, cell death occurs mainly in the progeny of High mis-segregation mitosis, whereas in PLK4OE considerable levels of cell death are also observed following other mitotic behaviors (Fig 2A, 2B and 2F, horizontal axis). In particular, around 35% of cell death was observed for the progeny of cells that had No mis-segregation or Slight mis-segregation defects, in contrast with PLK4Ctl where less than 10% of these cells died (Fig 2F, horizontal axis). In order to ascertain that our low spatio-temporal resolution imaging set-up is not hindering the detection of mitotic defects in PLK4OE that might explain this increased cell death, we performed live-imaging of Carboplatin-treated cells at higher spatiotemporal resolution. This imaging was performed during the first 24 h of Carboplatin treatment during which No mis-segregation or Slight mis-segregation events occur (Fig 2G). These events remained the dominating population observed in both PLK4Ctl and PLK4OE, and High mis-segregation of chromosomes remained infrequent in the first 24 h of Carboplatin treatment and at comparable levels in PLK4OE and PLK4Ctl (S2F Fig). We were able to determine the proportions of lagging chromosomes, chromatin bridges, and misaligned chromosomes and observed that PLK4OE mainly induces a nonsignificant tendency to increase chromosome mis-alignment compared to PLK4Ctl (S2G Fig). However, this behavior is also frequent in untreated PLK4OE cells where little cell death is observed (Fig 2D) suggesting it is not responsible for the observed increase in cell death in response to Carboplatin. Since the expected consequences of PLK4OE in cells is to promote multipolarity during mitosis, which may not have been detected at the low-resolution movies or in fixed preparations, we investigate whether multipolarity was detected in cells with extra centrosomes treated with carboplatin or DMSO. To do so, we generated iOVCAR8 cells expressing GFP-Tubulin and H2B-RFP to follow concomitantly mitotic spindle and chromosome behavior. Imaging cells at higher resolution and at short time intervals in time-lapse approaches revealed however, that the large majority of both PLK4OE and PLK4OE carboplatin-treated cells divide in a bipolar manner (S2H and S2I Fig). As seen in fixed cells, bipolar spindles contain clustered and un-clustered centrosomes and divide with/without chromosome mis-segregation errors (S2H Fig, top-second and third rows). Multipolar divisions are infrequent and interesting even presented centrosome clustering of certain extra centrosomes (S2H Fig, fourth row). It is worth noticing that certain cell divisions were extremely long, which may also be influenced by the co-expression of GFP-tubulin and H2B. Finally, an increase in cell death is observed in this cell line, suggesting that GFP-Tubulin expression may sensitize these cells to the effects of centrosome amplification and carboplatin (S2H Fig, fifth row and S2I Fig).

The temporal pattern of mitotic phenotypes with High mis-segregation events occurring when mitosis is initiated after a longer time spent in Carboplatin (Fig 2A and 2G) suggested an influence of the cell cycle on Carboplatin responses. We thus analyzed the cell cycle in response to Carboplatin using a FUCCI expressing OVCAR8 cell line. This strategy allowed us to discriminate cells in G1 from cells in S/G2 (Figs 2H, 2I, S3A and S3B), while still observing

cell death events which occur mainly in S/G2 (S3C Fig). First, we observed that Carboplatin induced an increase in S/G2 length, suggesting DNA damage in OVCAR8 cells activates the intra S and/or the G2/M checkpoint (S3D Fig), while G1 length was not strongly varying in any observed condition (S3E Fig). Next, we observed that S/G2 lengthening and subsequent delayed mitotic entry occurred mainly in cells that were in G1 at the onset of Carboplatin exposure, and therefore in cells exposed to Carboplatin for a complete S-phase (Fig 2H and 2J). Despite this delay, these cells most likely entered mitosis with unrepaired damage, driving High mis-segregation of chromosomes during mitosis and eventually leading to death of the progeny. In PLK4OE cells exposed to Carboplatin, the timing and proportions of mitotic and cell cycle behaviors was similar to PLK4Ctl. High mis-segregation events occurred in cells with a strong delay in mitotic entry (Fig 2B, 2F and 2G), which concurred with cells that were in G1 at the onset of Carboplatin exposure (Fig 2I and 2J). We did note a slight lengthening in G1 duration in Carboplatin-treated PLK4OE cells compared to PLK4Ctl (S3E Fig), but we did not observe a significant association between cell death and G1 length (S3F Fig).

These observations therefore suggest that PLK4Ctl cells are essentially killed by catastrophic mitosis induced by high levels of DNA damage and subsequent High mis-segregation of chromosomes. PLK4OE cells on the other hand also die following mitosis characterized by No or Slight mis-segregation of chromosomes occurring early upon Carboplatin treatment, suggesting they die independently of lethal mitotic behaviors. In agreement with cell death occurring independently of mitosis in PLK4OE, 17% cells died in the first generation compared to only 6% in PLK4Ctl (Fig 2A, 2B and 2E). In particular, cells that were in G1 at the beginning of Carboplatin exposure and therefore can accumulate DNA damage in their first cell cycle, were preferentially killed in PLK4OE with 25% of cells dying compared to 8% in PLK4Ctl (Fig 2H, 2I and 2K). Our findings propose that centrosome amplification sensitizes cells to the effect of Carboplatin in a single cell cycle and independently of catastrophic mitotic behaviors.

The centrosome has previously been involved in regulating the DNA damage response via recruitment of ATR, ATM, Chk1, and Chk2 [31,32]. We investigated whether centrosome amplification modifies the signaling downstream of DNA damage, explaining increased cell death in response to Carboplatin. We detected phosphorylation and activation of Chk1 and p53 in response to Carboplatin, but no difference is observed in PLK4OE compared to PLK4Ctl cell extracts (S3G and S3H Fig). In agreement with DNA damage levels and response being unchanged by PLK4OE, we also observed no difference in the levels of staining for the early marker of DNA damage response γ-H2AX, or for the recruitment of DNA damage repair factors FANCD2, 53BP1, and Rad51 (S3I and S3J Fig). Together, these observations suggest that centrosome amplification increases cell death in response to DNA damage, independently of catastrophic mitosis, but also independently of the DNA damage response.

## Centrosome amplification modulates mitochondrial apoptosis independently of p53 and the PIDDosome

In order to better understand how centrosome amplification increases cell death in response to Carboplatin independently of mitotic errors, we characterized the type of cell death and the associated signaling network. We observed that Carboplatin treatment induced apoptosis characterized by Caspase-3 cleavage (Fig 3A and 3B), as well as cells becoming positive for Annexin-V by flow cytometry, which was completely suppressed by the pan-caspase inhibitor Q-VD-Oph (Fig 3C). PLK4OE induces both a premature cleavage of Caspase-3 (t = 48 h compared to t = 72 h in PLK4Ctl), as well as an increase in the Annexin-V positive cell population (53% compared to 32% in PLK4Ctl), confirming that centrosome amplification potentiates the apoptotic response to Carboplatin (Fig 3A–3C). To determine if the MOMP-dependent

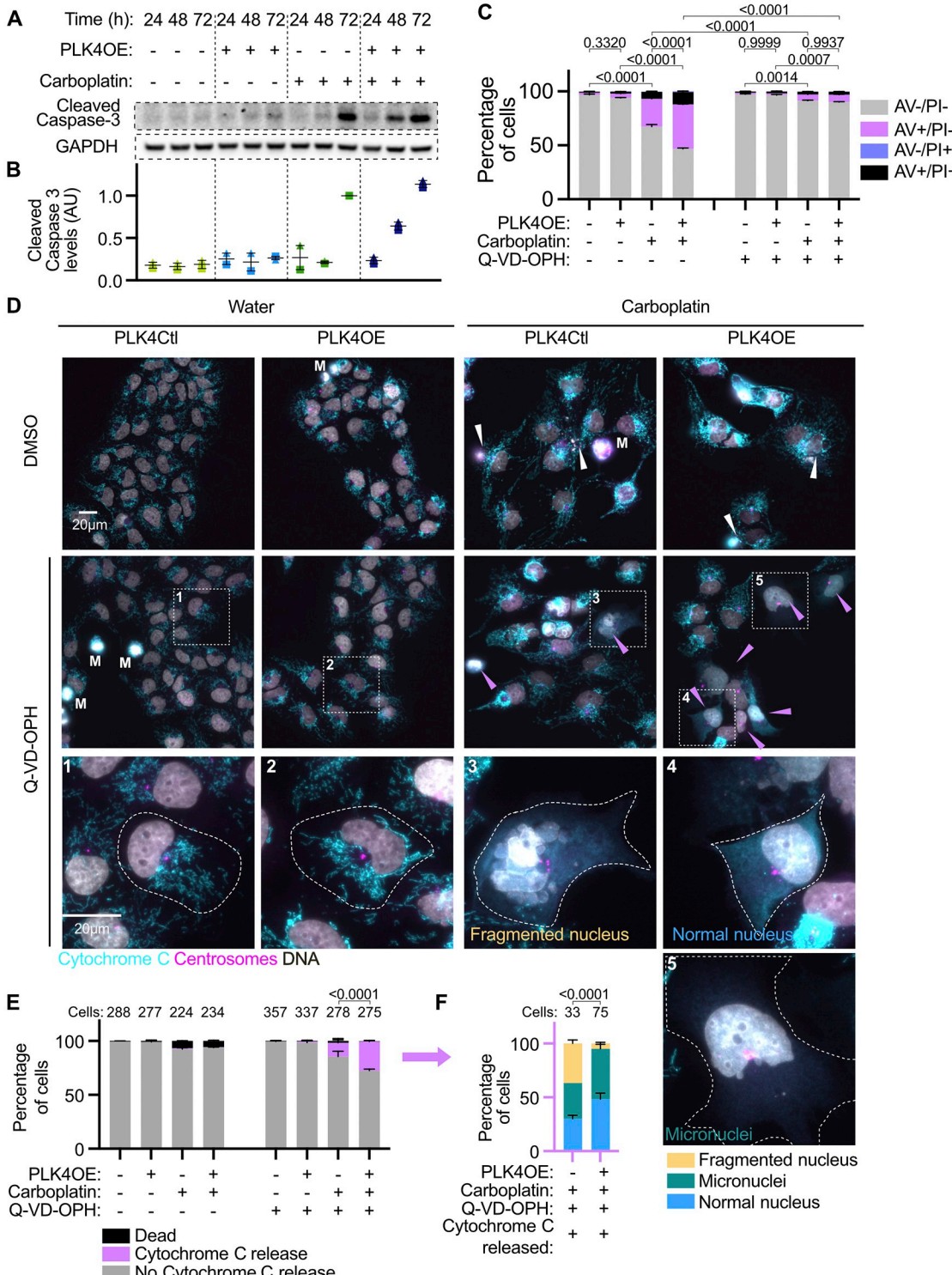

**Fig 3. Centrosome amplification enhances mitochondrial outer membrane permeabilization in response to Carboplatin. (A)** Representative western blot detecting Caspase-3 cleavage. **(B)** Average and SEM of cleaved caspase 3 protein levels from 2 independent experiments, normalized to levels measured in Carboplatin treated PLK4Ctl cells at 72 h. **(C)** Bar graphs showing the average and SEM of the percentage of cells in specified Annexin V-APC/PI gates analyzed by flow cytometry. Four replicates obtained from 2 independent experiments, with a minimum of 60,000 cells analyzed per condition and replicate. Statistical test: comparison of the percentage of Annexin V positive cells, using ANOVA with Sidak's multiple

comparison test. Representative cytometry profiles can be found in the S1_Appendix. (**D**) Representative images of cells labeled with DAPI (gray) and antibodies against Cytochrome C (cyan) and centrosomes in pink. White arrows indicate dead cell debris, pink arrows indicate cells that have released Cytochrome C in the cytoplasm, M indicates mitotic cells. Representative insets are shown for individual Q-VD-Oph treated cells. (**E**) Bar graphs showing the average and SEM of the percentages of indicated cell populations. Two independent experiments, statistical test: Fisher's exact test on the number of cells releasing Cytochrome C. Numbers on the top of each graph represent the number of cells analyzed per condition. (**F**) Left: Bar graphs showing the average and SEM of the percentages of cells with indicated nuclear phenotypes, within the populations releasing Cytochrome C. Two independent experiments, statistical test: Fisher's exact test on the number of cells with fragmented nuclei. Numbers on the top of each graph represent the number of cells analyzed per condition. Right: Representative images of DAPI-stained nuclei with indicated phenotypes. Data for Fig 3 can be found in S3 Data.

apoptotic pathway was activated in response to Carboplatin, we stained cells for Cytochrome C in order to observe its release from mitochondria (Fig 3D). In untreated cells, Cytochrome C was detected in punctate structures throughout the cytoplasm, in agreement with its mitochondrial localization. Upon Carboplatin exposure, Cytochrome C staining remained similar to untreated cells, while we observed dead cell remnants characterized by condensed DNA (Fig 3D, white arrows and 3E). However, if release of Cytochrome C occurred, it could lead to immediate apoptosis initiation and detachment of the cells, precluding the observation of cells via immunofluorescence. We ctherefore used the pan-caspase inhibitor Q-VD-Oph to inhibit the effector steps of apoptosis downstream of Cytochrome C release, and observed a population of cells where Cytochrome C is diffuse in the cytoplasm and nucleus (Fig 3D, pink arrows and 3E). MOMP, Cytochrome C release and caspase activation therefore occurs in response to Carboplatin, suggesting that mitochondrial apoptosis is the main cell death mechanism at play.

Importantly, we observed 27% of cells releasing Cytochrome C in PLK4OE cells compared to only 12% in PLK4Ctl cells, suggesting that Carboplatin induces a stronger activation of apoptosis in the presence of centrosome amplification (Fig 3D and 3E). In agreement with different stresses leading to apoptosis between PLK4Ctl and PLK4OE, we observed that 36% of the cells that released Cytochrome C presented fragmented nuclei in PLK4Ctl compared to 5% in PLK4OE (Fig 3D and 3F). These fragmented nuclei are symptomatic of high chromosome mis-segregation during mitosis, supported by PLK4Ctl cells being killed by catastrophic mitosis, while centrosome amplification potentiates apoptosis independently of catastrophic mitosis in PLK4OE.

The canonical intrinsic apoptosis pathway linking the DNA damage response to MOMP occurs via p53 stabilization which then drives the transcription of pro-apoptotic BCL2 family genes [38]. Centrosome amplification has been linked to p53 stabilization via PIDDosome activation, which is dependent on centriole distal appendage grouping. This leads to Caspase-2 cleavage and activation and cleavage of MDM2—a major p53 regulator—[34,35]. OVCAR8 cells have a mutant TP53 gene which leads to alternative splicing of exon5, and a 6 amino-acid deletion in p53's DNA-binding domain [39]. We observed that p53 protein is present, phosphorylated, and accumulates in response to Carboplatin (S3G Fig, quantified in S4B Fig). The deletion in p53's DNA damage binding domain is suggested to preclude its transcriptional activities [39], and in agreement with this, we observed that its transcriptional targets p21 and PUMA do not show a striking increase upon Carboplatin exposure (S4A and S4B Fig). We nevertheless assessed p53's involvement in Carboplatin-induced cell death using shRNA, and showed that p53 was dispensable in both PLK4Ctl and PLK4OE in response to Carboplatin (S4C and S4D Fig). p21 is best characterized for its functions in cell cycle arrest and apoptosis inhibition, but has also been shown to up-regulate apoptosis [40]. As we noticed p21 to be up-regulated in PLK4OE cells compared to PLK4Ctl (S4A and S4B Fig), we tested its contribution to cell death, but observed no effect of knocking down the p21 coding gene CDKN1a (S4C and S4D Fig). Despite apoptosis being p53 independent in Carboplatin treated OVCAR8, we tested whether PIDDosome activation may contribute to enhancing intrinsic apoptosis in PLK4OE. Indeed, upon PLK4OE, we observed cleavage of Caspase-2 and MDM2 reflecting PIDDosome

activation (S4E and S4F Fig). We knocked-down the distal appendage protein required for PID-Dosome activation ANKRD26 and observed a strong reduction of Caspase-2 and MDM2 cleavage in PLK4OE, reflecting efficient PIDDosome silencing (S4G Fig). However, this had no effect on the enhanced cell death observed upon Carboplatin exposure in PLK4OE, suggesting that centrosome amplification potentiates apoptosis independently of the PIDDosome (S4H Fig). Altogether, our results show that centrosome amplification leads to enhanced apoptosis in response to Carboplatin independent of previously described centrosome signaling nodes.

## Centrosome amplification primes for MOMP and sensitizes cells to a diversity of chemotherapies

So far we have observed that PLK4OE cells execute apoptosis faster and to a higher proportion, in response to a level of stress (mitotic behaviors, DNA damage levels and DNA damage response) which is no different than in PLK4Ctl. This therefore suggested that these cells may be primed for MOMP, meaning that the balance between pro-apoptotic and anti-apoptotic BCL2 family proteins that determine the activity of the mitochondrial pore forming proteins BAX and BAK is tilted towards their activation in PLK4OE [41]. To test this possibility, we performed MTT dose-response assays during 72 h to drugs which mimic the activity of specific pro-apoptotic BH3-only BCL2 family proteins. Strikingly, we observed that PLK4OE induces a strong sensitization to WEHI-539—a specific BCL-XL inhibitor—(Fig 4A and 4B), but not to Venetoclax or A1210477—specific inhibitors of BCL-2 and MCL-1, respectively—(S5A and S5B Fig). We confirmed that 72 h 300 nM WEHI-539 exposure selectively induced apoptosis in PLK4OE cells in a Caspase dependent manner, via Annexin V and PI cytometry (Fig 4C). We also observed release of Cytochrome C from mitochondria in WEHI-539-treated PLK4OE cells upon pan-caspase inhibition, confirming that BCL-XL inhibition induces MOMP specifically in PLK4OE (Fig 4D and 4E). Counting centrosomes in PLK4OE cells revealed that the 72 h 300 nM WEHI-539 treatment reduced the proportion of cells with extra centrosomes to the level observed in PLK4Ctl cells (Fig 4F and 4G). These results suggest efficient killing of cells with centrosome amplification, which is suppressed upon pan-caspase inhibition. We were also able to count centrosomes in the population of cells that release Cytochrome C from mitochondria and are therefore poised to die, revealing that the majority of these cells have centrosome amplification in WEHI-539 treated PLK4OE cells (Fig 4F). Interestingly, an extremely small proportion (1%) of PLK4Ctl cells also release Cytochrome C in response to WEHI-539 (Fig 4E) and counting centrosomes in these cells revealed a higher level of centrosome amplification than the untreated PLK4Ctl population (Fig 4F). This suggests that independently of induced PLK4 overexpression, centrosome amplification primes for MOMP in OVCAR8 cells. Unfortunately, SAS-6 overexpression as an alternative mean to induce centrosome amplification only lead to a weak increase in the number of cells with more than 2 centrosomes and we therefore did not test if apoptotic priming occurs in this system (S5C Fig). However, we observed that in parental OVCAR8 cells devoid of the inducible PLK4OE transgene, cells with supernumerary centrosomes are preferentially killed by WEHI-539 suggesting this apoptotic sensitization occurs in cells with "naturally" occurring centrosome amplification (S5D Fig).

We next tested if similar effects might be observed in other cell lines and established inducible centrosome amplification via PLK4OE in ovarian cancer cell lines COV504 and SKOV3 (S5E and S5F Fig). We tested if MOMP priming identified in OVCAR8 was also observed in these cell lines (S5G Fig). We used the less specific BH3-mimetic Navitoclax (inhibitor of BCL2, BCL-XL, and BCL-W), as the dependency on BCL-XL in PLK4OE OVCAR8 might be reflecting OVCAR8 apoptotic wiring rather than a specific effect of centrosome amplification on BCL-XL. We observed that Navitoclax reduced the viability of PLK4OE cells preferentially

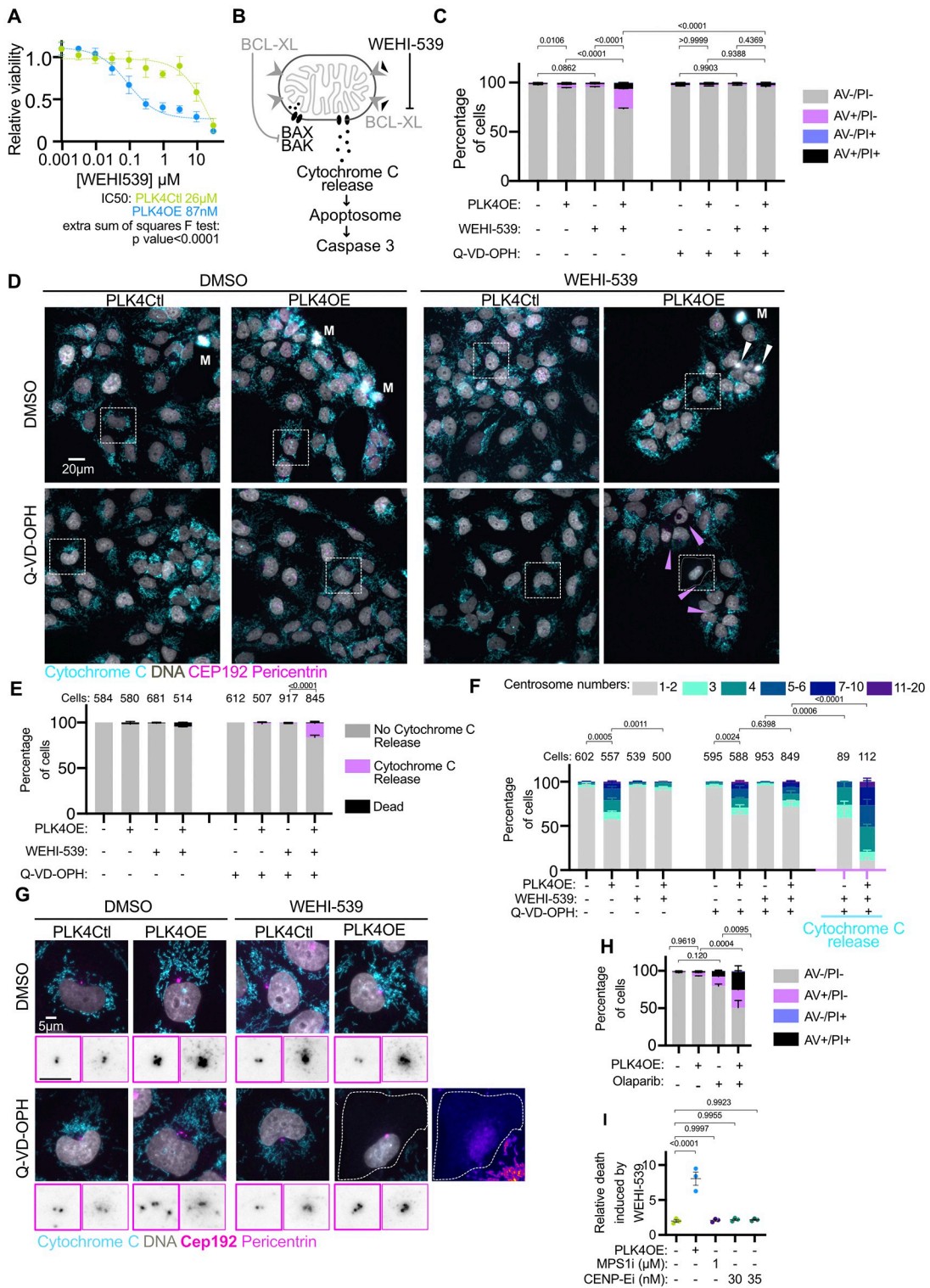

**Fig 4. Centrosome amplification primes for mitochondria outer membrane permeabilization. (A)** Dose-response of PLK4Ctl and PLK4OE cells to WEHI-539 normalized to their respective untreated conditions, obtained from MTT viability assays. Mean and SEM of 3 independent experiments each obtained from averaging 3 technical replicates. **(B)** Schematic of the induction of apoptosis by WEHI-

539. When BAX and BAK channel formation are inhibited by BCL-XL, Cytochrome C is present in the mitochondria intermembrane space (left). WEHI-539 inhibits BCL-XL which relieves the inhibition of channel formation by BAX and BAK, leading to Cytochrome C release, Apoptosome activation, and cleavage of Caspase 3 (right). **(C)** Bar graphs showing the average and SEM of the percentage of cells in specified Annexin V-APC/PI gates analyzed by flow cytometry. Four replicates obtained from 2 independent experiments with a minimum of 15,000 cells analyzed per condition and replicate. Statistical test: comparison of the percentage of Annexin V positive cells, using ANOVA with Sidak's multiple comparison test. Representative cytometry profiles can be found in the S1 Appendix. **(D)** Representative images of cells labeled with DAPI (gray) and antibodies against Cytochrome C (cyan), CEP192 (magenta), and Pericentrin (magenta). White arrows indicate dead cell debris, pink arrows indicate cells that have released Cytochrome C in to the cytoplasm, M indicates mitotic cells. Representative insets are shown in panel (G). **(E)** Bar graphs showing the average and SEM of the percentages of indicated cell populations. Two independent experiments, statistical test: Fisher's exact test on the number of cells releasing Cytochrome C. Numbers on the top of each graph represent the number of cells analyzed per condition. **(F)** Bar graphs showing the average and SEM of the percentage of cells with the indicated number of centrosomes (determined by the co-localization of CEP192 and Pericentrin). Two independent experiments, statistical test: comparison of the percentage of cells with more than 2 centrosomes, using ANOVA with Sidak's multiple comparison test. Numbers on the top of each graph represent the number of cells analyzed per condition. **(G)** Insets from panel (D) with inverted grayscale insets zooming on the centrosomes showing CEP192 (tick magenta border) and Pericentrin (light magenta border). For PLK4OE, WEHI-539, Q-VD-OPH a "Fire-lut" inset is shown. **(H)** Bar graphs showing the average and SEM of the percentage of cells in specified Annexin V-APC/PI gates analyzed by flow cytometry. Four replicates obtained from 2 independent experiments with a minimum of 10,000 cells analyzed per condition and replicate. Statistical test: comparison of the percentage of Annexin V positive cells, using ANOVA with Sidak's multiple comparison test. Representative cytometry profiles can be found in the S1 Appendix. **(I)** Scatter dot plots showing the ratio between the percentages of Annexin V positive cells observed in presence and absence of WEHI-539 300 nM. Average and SEM of 3 replicates from 2 independent experiments. Statistical test: ANOVA with Dunnett's multiple comparison test. Data for Fig 4 can be found in S4 Data.

compared to PLK4Ctl cells in COV504 (EC80 of 400 nM and 3,6 μm, respectively) although to a lesser extent than in OVCAR8 (EC80 of 40 nM and 2,2 μm, respectively). Priming was however not observed in SKOV3. We then established the IC50s for Carboplatin and Paclitaxel, and determined combination concentrations in PLK4Ctl cells (S5H Fig). We used Trypan blue assays to determine viability in response to chemotherapy and confirmed this approach in OVCAR8 by showing that viability decreases more in PLK4OE compared to PLK4Ctl (S5I Fig, top). Interestingly, we observed a gradation in the enhanced cell death induced by centrosome amplification in the different cell lines, with the strongest effect observed in OVCAR8 (S5I Fig, top), the weakest in SKOV3 (S5I Fig, bottom), and an intermediate effect in COV504 (S5I Fig, middle), which can be put in perspective with the observed gradation in MOMP priming. For responses to Paclitaxel however we have to take into account the fact that multipolar divisions are also increased in COV504 PLK4OE cells (S5J Fig).

Our identification of apoptotic priming in cells with centrosome amplification suggests that it might be associated with enhanced cell death in response to a larger panel of drugs. We therefore tested whether centrosome amplification sensitizes ovarian cancer cells to PARP inhibitors which are now included in standard of care protocols in epithelial ovarian cancer [42], focusing on Olaparib (IC50s determined and presented in S5H Fig). Trypan Blue viability assays in OVCAR8 and COV504 confirmed that PLK4OE leads to reduced viability compared to PLK4Ctl in response to Olaparib (S5K Fig). This effect was not observed in SKOV3 in which we have not observed MOMP priming linked to centrosome amplification. Using flow cytometry to detect Annexin V positive cells, we confirmed that in OVCAR8 Olaparib indeed induces more apoptosis in PLK4OE than PLK4Ctl (Fig 4H). Together, our results suggest that centrosome amplification enhances cell death in response to chemotherapy differentially depending on the cell line and that centrosome amplification associated apoptotic priming can sensitize to a diversity of chemotherapies.

## Centrosome amplification primes for MOMP independently of chromosome instability or lengthened mitosis

Centrosome amplification induces an increase in chromosome instability and a spindle-assembly checkpoint-dependent extension of mitosis duration [7,8,11], both of which we

observed in PLK4OE OVCAR8 cells (S1E and S1G Fig). Apoptotic priming and in particular sensitization to BCL-XL inhibition has previously been linked to mitotic defects. In particular chromosome instability, micronuclei formation and cGAS/STING signaling can drive a transcriptional response that drives apoptosis or priming [43,44]. Alternatively, extended mitotic duration can lead to the proteosomal degradation of anti-apoptotic BCL2 family proteins, leading to BCL-XL sensitization [45–48]. We were therefore interested in determining if apoptotic priming observed in response to centrosome amplification is caused by cumulated mitotic stress in these cells.

First, we aimed to better characterize the mitotic stress induced by centrosome amplification in the already chromosomally instable OVCAR8 cell line (S1G Fig). We used single-cell DNA-sequencing to assess karyotype heterogeneity and observed scores of 0,119 in PLK4Ctl, 0,137 in PLK4OE cells, and 0,283 in PLK4Ctl cells treated with 1 µm of the MPS1 inhibitor AZ3146 as a positive control of chromosome mis-segregation (S6A Fig). PLK4OE therefore only mildly increased aneuploidy, in line with levels of chromosome-mis-segregation observed by time-lapse imaging (S6B Fig). We also assessed the extent of mitotic lengthening induced in PLK4OE cells (S6C Fig) and observed it was mild (median = 60 min in PLK4OE and median = 35 min in PLK4Ctl) compared to that induced by low doses of the CENP-E inhibitor GSK923295 (median = 100 min at 30 nM and 175 min at 35 nM), despite levels of chromosome mis-segregation being similar (S6B Fig).

Mitotic stress is therefore mild in PLK4OE compared to the other perturbations we tested, but we were nevertheless interested in determining if it contributes to apoptotic priming. We were not able to reduce mitotic duration in PLK4OE cells via spindle assembly checkpoint inhibition without inducing a strong increase in multipolar divisions, so we used MPS1 and CENP-E inhibition to mimic mitotic stress observed in PLK4OE. We pretreated PLK4Ctl cells with inhibitors during 72 h before adding WEHI-539 for an additional 24 h. In PLK4OE cells, this lead to 33% Annexin V positive cells, whereas it only induces 9% in response to MPS1 inhibition (S6D Fig), making it unlikely that priming occurs in response to chromosome instability in PLK4OE. In CENP-E inhibition pretreated cells at 30 nM and 35 nM, WEHI-539 induces 12% and 25% Annexin V cells, respectively, in line with mitotic lengthening inducing priming [48]. Importantly however, 35 nM CENP-E inhibition pretreatment already induces 12% Annexin V positive cells which is considerable compared to the 4% observed in PLK4OE (S6D Fig), and most likely is explained by the extensive mitotic lengthening observed in response to 35 nM CENP-E inhibition (S6C Fig). This results in the ratio of cell death induced by WEHI-539 relative to the basal observed level of cell death to be comparable in PLK4Ctl cells and CENP-E inhibition pretreated cells (around 2-fold). In contrast, the ratio of cell death was much higher and close to 8-fold increase upon PLK4OE (Fig 4I). Therefore, the priming induced in PLK4OE stands out from that induced by other sources of mitotic stress in that PLK4OE cells are viable but strongly dependent on BCL-XL. These results also suggest that the combination of chromosome instability and mitotic lengthening is not the major contributor to MOMP priming upon centrosome amplification in OVCAR8 cells.

To identify transcriptomic signatures that may influence cell death responses in cells with extra centrosomes, we used bulk RNAseq comparing PLK4OE and PLK4Ctl OVCAR8 cells. A strong inflammatory signature in PLK4OE (S6E Fig) was identified, and we also observed STING phosphorylation (S6F Fig), suggesting that the cGAS/STING pathway may shape the transcriptional response to centrosome amplification. We were therefore interested in directly testing if cGAS/STING signaling might contribute to priming, although this seemed unlikely as CENP-E and MPS1 inhibition also activate STING (S6F Fig) but are not associated with priming. We used a bulk LentiCRISPR knock-out approach of STING, but observed no influence on PLK4OE cells sensitivity to WEHI-539 (S6G and S6H Fig).

We therefore identify that centrosome amplification in OVCAR8 leads to MOMP priming which is revealed by a selective sensitization to the BCL-XL inhibitor WEHI-539. Comparing with other mitotic perturbations, we conclude that the centrosome amplification associated priming is independent of mitotic lengthening and chromosome instability.

## High centrosome numbers are associated with a better response to chemotherapy in a high grade serous ovarian cancer patient cohort

To assess if centrosome amplification is associated with chemotherapy responses in patients, we turned to the characterization of centrosomes we previously performed in situ in treatment-naive epithelial ovarian tumors (Cohort description in S1 Table). Here, centrosomes were detected as the colocalization of Pericentrin and CDK5RAP2 in confocal images of methanol-fixed patient tissue sections [5]. To assess centrosome numbers in samples we defined the centrosome to nucleus ratio (CNR) as the number of centrosomes detected in a field by the number of nuclei which we averaged over 10 fields per patient (S7A Fig). In healthy tissues obtained from prophylactic oophorectomy or hysterectomy, the CNR was 1.02 ± 0.02 suggesting cells have on average 1 centrosome per cell which is expected for a non-proliferative tissue. On average, the CNR in tumor tissues was 1.43 ± 0.04, with the minimum at 0.61 and maximum at 2.55. While only 9% of tumors had a CNR above 2, suggesting that pervasive centrosome amplification—when defined by the presence of more than 2 centrosomes per cell—is infrequent, 89% of the tumors presented a CNR superior to the average CNR found in healthy tissues. Considering that the CNR did not correlate with proliferation as established by the mitotic index (S7B Fig) and Ki67 staining (S7C Fig), centrosome amplification could contribute to this increase in CNR in tumors compared to healthy samples. Indeed, we did observe nuclei associated with more than 2 centrosomes, although the tumor-wide frequency of such events reached a maximum of 3,2% in our previous study [5].

We next examined if the CNR was associated with chemotherapy responses restricting our analysis to the high-grade serous ovarian cancers (HGSOCs) in our cohort. We dichotomized our population into 2 groups using the Classification and Regression Trees (CART) method. Taking into consideration the binary outcome "relapse within 6 months or no relapse within 6 months," this method resulted in the categorization of the cohort into 55 low CNR (≤1.45) and 33 high CNR (>1.45). Importantly, we observed no association between CNR and FIGO stage. Most patients in this cohort—59%—are stage III patients and comprise both high and low CNRs (S7D Fig). We next plotted HGSOC patient survival curves according to the CNR status. We found that high CNR was associated with better overall survival (S7E Fig). These results suggest that despite its oncogenic potential [7,17–19], centrosome amplification might improve patient prognosis at least in ovarian cancer. This puzzling observation could be explained if high CNR promotes chemotherapy responses as overall survival data reflects patients complete clinical course which includes Carboplatin and/or Paclitaxel therapy for 84% of patients in this cohort. To directly assess a link between CNR and chemotherapy responses, we plotted patient time to relapse and found that high CNR was associated with a longer time to relapse (S7F Fig). Together, this work suggests that centrosome status in ovarian cancer can influence patient outcome, in particular with high CNR potentiating the response to chemotherapy. The overall low levels of centrosome amplification observed in the patients nevertheless ask if and how the preferential targeting of these cells can drive an improved therapeutic response, prompting the need for in vivo experimentation.

## Discussion

Centrosome amplification as a therapeutic target has been mainly explored from the prism of multipolar division and mitotic drugs such as HSET or Aurora A inhibitors have the potential to induce multipolar divisions and so to efficiently target cancer cells in vitro [22,25], even if the potential of mitotic inhibitors have so far been unsuccessful in clinical trials [49]. Our results identify apoptotic priming as a novel cell death susceptibility conferred by centrosome amplification. In particular, we show that centrosome amplification sensitizes cells to BH3-mimetic drugs.

The apoptotic priming seems to be specific to centrosome amplification, rather than a consequence of the associated mitotic stress. Possible causes of this priming could be disruption of mitochondrial networks during mitosis, or in interphase. This may be in link with recent observations of subcellular reorganization in response to centrosome amplification in RPE-1 cells [50]. Centrosomes are also involved in multiple signaling pathways [32,51] and given the pleiotropic effects of centrosome amplification which also induce ROS and inflammation [15]; we consider that the best method to identify the source of the priming would be whole-genome screening approaches.

From a clinical perspective, our analysis of a patient cohort shows that high centrosome numbers limit relapse in response to chemotherapy, indicating that centrosome amplification must be considered beyond its malignant potential. Given the toxicity of cytotoxic therapies, the perspective of better patient stratification and response prediction, considering centrosome amplification as a sensitizing factor offers promising perspectives. Our observation that centrosome amplification enhances cell death independently of multipolar mitosis broadens the therapeutic importance of this cancer cell feature, beyond treatments that target spindle assembly and mitosis.

Our identification of apoptotic priming in cells with centrosome amplification also has clinical relevance, relative to the use of BH3-mimetics in the clinic. Multiple clinical trials involve the use of these drugs alone or in combination with conventional chemotherapy, with the most promising example being BCL-2 inhibitor Venetoclax in combination with Azacitidine, which is approved for the treatment of acute myeloid lymphoma. However, BH3 mimetic use in the clinic is hampered by lack of good prognostic markers and on-target toxicity of BCL-XL inhibitors WEHI-539 and Navitoclax, leading to thrombocytopenia [52]. Our findings are interesting for both of these challenges, first by placing centrosome amplification as a potential biomarker for sensitivity to BH3-mimetics. Additionally, we ask if the apoptotic priming we describe could be related to BCL-XL inhibitor toxicity, via sensitization of platelet-producing megakaryocytes that present centrosome amplification [53].

There are multiple limitations to our study. We must emphasize that the levels of centrosome amplification in the cohort is low [5] and that centrosome loss might also contribute to modulating chemotherapy responses. It nevertheless remains interesting to consider that targeting low levels of centrosome amplification could have an observable clinical effect, and to explain these results we propose that elimination of cells with centrosome amplification might be advantageous given the malignant potential of these cells [7,13–15,19]. We are also eager to know if centrosome numbers influence responses to chemotherapy in additional epithelial ovarian cancer cohorts and in different cancer types. An important step to facilitate broader studies is the automatization of centrosome detection and counting in patient tissues. Additionally our identification of apoptotic priming in response to centrosome amplification and the associated clinical perspectives justify the need for a better understanding of the priming mechanism induced by centrosome amplification. This would also help identify the contexts in which this priming emerges as we have not observed it in all the cancer cell lines studied.

## Material and methods

### Study design

This work is a study of the influence of centrosome amplification on the response to chemotherapy in epithelial ovarian cancer. The objectives of the cell biology work were to identify if and how centrosome amplification potentiates cell death in response to chemotherapy, using a combination of single-cell live imaging, and classical cell biology experiments such as cytometry and western blot. All the presented data has been replicated in 2 to 5 biological replicates. The objective of the clinical work was to identify if centrosome numbers influence clinical parameters in a patient cohort. All samples were taken before chemotherapy administration and obtained from the Biological Resource Center (BRC) of Institut Curie (certification number: 2009/33837.4; AFNOR NF S 96 900). Patients provided oral consent for their samples to be used for research purposes. In compliance with the French regulation, patients were informed of the studies performed on tissue specimens and did not express opposition. The National Commission for Data Processing and Liberties (N˚ approval: 1487390) approved all analysis, as well as The Institutional Review Board and Ethics committee of the Institut Curie. Centrosomes were previously stained and detected in an HGSOC patient cohort of 100 patients [5]. Here, we determined an index allowing the quantitative assessment of centrosome numbers in patient tissues, which we then correlated with patient clinical parameters. Data collection for each experiment is detailed in the respective figure legend.

### Cell lines and cell culture

All cell lines were cultured at 37˚C with 5% CO2 in DMEM/F12 media (Thermo Fisher Scientific #31331028) supplemented with 10% Tetracyclin-free Fetal Bovine Serum (Dutscher #500101L), 100 μg/ml streptomycin, and 100 U/ml penicillin (Thermo Fisher Scientific #15140122). OVCAR8 and COV504 were obtained from the laboratory of F. Mechta-Grigoriou, and SKOV3 were purchased from ATCC (#HTB-77). Cell cultures underwent authentication by short tandem repeat analysis (powerplex16 HS kit, Promega #DC2101) and were routinely checked for mycoplasma (PlasmoTest Mycoplasma detection kit, InvivoGen, #rep-pt1).

### Cell line generation

Inducible PLK4 overexpression, H2B-RFP expression, FUCCI expression, GFP-tubulin expression, shRNA expression, and bulk CRISPR-Cas9 knock-out of STING were stably established by lentiviral infection. Viruses were produced in HEK cells using Lipofectamine 2000 (Thermo Fisher Scientific #11668019) to co-transfect lentiviral constructs with pMD2.G and psPAX2 plasmids. Viral particles were collected in the supernatant 48 h after transfection, filtered, and used to infect the cell lines during 24 h. Cells were then FACS sorted (inducible PLK4 overexpression, H2B-RFP, Tubulin-GFP) or selected using Puromycin at 5 μg/ml (shRNA lines and CRISPR-Cas9 knock-out of STING) or using Blasticidin at 5 μg/ml (when the plasmid Lenti Tet-ON Myc-hPLK4 Blasticidin is used to induce PLK4 overexpression). Efficiency of knock-down and knock-out was assessed by western blot. The list of plasmids used is available in S2 Table.

### Drug treatments

All chemicals used are listed in S3 Table. To induce centrosome amplification, cells were exposed to doxycycline (1 μg/ml) or DMSO (diluent control, 1/10,000) for 72 h. If cells were subsequently treated with another drug, cells were detached and replated without addition of

doxycycline to the PLK4OE population, and left to attach for 8 h. Drug treatments were then carried out for 72 h at the indicated concentrations. For the experiments comparing centrosome amplification to other mitotic stresses (CENP-E and MPS1 inhibition), cells were exposed to doxycycline (1 μg/ml for centrosome amplification), AZ3146 (1 μm, for MPS1 inhibition), GSK923295 (30 to 35 nM, for CENP-E inhibition), or DMSO (diluent control) for 72 h. Subsequent treatments (WEHI-539) or analysis (live-imaging of mitotic phenotypes) were then carried out in presence of the same initial concentrations of drug for 24 h.

## Live-imaging and analysis

For live-imaging of chemotherapy responses, cells were plated on Ibidi μ-Slide 8 Well slides (Clinisciences, #80806-G500). Chemotherapy treated and untreated cells from both PLK4Ctl and PLK4OE populations were imaged during the same experiment. Imaging was performed with a 20× dry objective (CFI Plan Apo LBDA 20× 0,75N.A) via an EMCCD camera (Evolve, Photometrics) on an inverted microscope (Inverted Ti-E Nikon) equipped with a spinning disk (CSU-X1 Yokogawa), a stage-top temperature and CO2 incubator (Tokai Hit) and integrated in Metamorph software. For each well, 4 to 6 positions were acquired every 10 min during 72 h, with a single slice in the brightfield channel and 10 3 μm slices per z-stack in the H2B-RFP channel or in the mKO2-Cdt1(30–120) and mAzami-Green-Gem1(1–110) channels for the FUCCI cells. For higher resolution imaging, cells were plated in 4 compartment Cellview dishes (Greiner Bio-One 627870) and 3 positions were acquired in each well every 2,5 min during 24 h, using a 40× oil objective (CFI Plan Fluor 40× 1,3N.A), acquiring a single slice in the brightfield channel and 20 1 μm slices per z-stack in the H2B-RFP channel. Time-lapse movies were then analyzed manually using a custom Fiji macro to record a list of events, and a custom Python script to generate excel data files and single-cell profiles.

For live-imaging of mitotic phenotypes induced by centrosome amplification, MPS1 inhibition and CENP-E inhibition, the same approach was used, acquiring each position every 5 min during 24 h. Imaging was performed using the equipment described above.

## Immunofluorescence

Cells were plated on 18-mm glass coverslips in 12-well plates. Cells were fixed for 5 min in ice-cold methanol (for S1A and S5E Figs), for 10 min in 4% PFA in PBS at room temperature (for Figs 3D, 4D and 4G), for 10 min in 4% PFA in PBS at 4˚C (for S3I Fig), or for 10 min in 4% PFA + 0,025% Glutaraldehyde in 80 mM Pipes + 5 mM EGTA + 1 mM MgCl2 + 0,1% Triton X-100, before 10 min quenching in 0,1% NaBH4 (for S1I, S2A and S2D Figs). Cells were washed 3 times in PBST (PBS + 0,1% Triton X-100) and incubated in PBST + BSA 0,5% for 30 min at room temperature. Cells were then incubated for 1 h in primary antibodies diluted in PBST + BSA 0,5%, washed 3 times in PBST, incubated for 30 min in secondary antibodies diluted in PBST + BSA 0,5%, and washed 3 times in PBST. Cells were then stained for DNA using 3 μg/ml DAPI diluted in PBST + BSA 0,5%, washed 3 times in PBS, and mounted with mounting medium (1.25% n-propyl gallate, 75% glycerol, in H2O). Antibodies used are listed in S4 Table.

## Immunofluorescence imaging and quantifications

Immunofluorescence images were acquired with a sCMOS camera (Flash 4.0 V2, Hamamatsu) on a widefield microscope (DM6B, Leica systems), with a 63× objective (63× HCX PL APO 1.40 to 0.60 Oil from Leica) or a 100× objective (100× HCX PL APO 1.40 to 0.70 Oil from Leica, for S1I, S2A and S2D Figs), using Metamorph software. Z-stacks were acquired at 0,3 μm intervals.

Centrosome numbers, cytochrome C release, and mitotic spindles were scored manually. DNA damage marker intensity or foci number were determined on z-projections of images,

using a custom Python script to run the h_maxima function from the skimage.morphology. extrema module.

## Western blotting

Cells were lysed in RIPA (150 mM sodium chloride, 1% NP-40, 0.5% sodium deoxycholate, 0.1% sodium dodecyl sulfate, 50 mM Tris, pH 8.0) complemented with protease (Sigma-Aldrich #11697498001) and phosphatase (Sigma-Aldrich #4906845001) inhibitors. Samples were dosed using a BiCinchoninic acid Assay (Pierce BCA protein assay, Thermo Fisher Scientific #23227). Samples were diluted in RIPA with 4X NuPage LDS sampling buffer (Thermo Fisher Scientific #NP0007) and heated at 80˚C for 10 min. Approximately 20 μg of protein was loaded in Bolt 4–12% Bis-Tris precast gels (Thermo Fisher Scientific #NW04125BOX) and subjected to electrophoresis in Bolt MOPS SDS running buffer (Thermo Fisher Scientific #B0001). The gels were transferred to nitrocellulose membranes (Dutscher #10600001) using transfer buffer (25 mM Tris, 192 mM Glycine, 20% Methanol) for 90 min at 4˚C. Membranes were stained in primary or horseradish peroxidase coupled secondary antibodies diluted in PBS or TBS + 0,5% Tween 20 + 0,5% BSA or non-fat milk according to providers instructions. Membranes were first stained using Ponceau, before saturating for 1 h at room temperature in 5% non-fat dry milk or 5% BSA in PBS or TBS + 0,5% Tween20. Membranes were then incubated overnight in primary antibodies, washed 5 times in PBS or TBS + 0,5% Tween 20, then incubated for 1 h at room temperature in secondary antibodies. Membranes were then washed again 5 times in PBS or TBS +0,5% Tween 20. Horseradish peroxidase reaction was developed using SuperSignal Plus Chemiluminescent substrates (Thermo Fisher Scientific #34580 and #34094) and imaged (BioRad ChemiDoc MP). The Image Lab software (BioRad version 6.0.1) was used to measure background-adjusted volume intensity, which was normalized using GAPDH signal. Antibodies used are listed in S4 Table.

## Transfection

HT-DNA was transfected as a positive control for cGAS/Sting activation. Transfection of 1 μg/ml HTDNA was carried out using Lipofectamine 2000 (Thermo Fisher Scientific #11668019) for 24 h.

## Cytometry

Cells were detached, rinsed in PBS, rinsed in AnnexinV Binding Buffer (BioLegend #422201), and around 100,000 cells were resuspended in 50 μl Annexin V Binding Buffer. Cells were stained using Annexin V APC and Propidium Iodide (Biolegend #640932) at 0,6 μg/ml and 10 mg/ml, respectively. Cells were incubated for 15 min, and then diluted in 200 μl Annexin V Binding Buffer. Cells were analyzed using a Bio-Rad ZE5 analyzer, and data was analyzed using FlowJo 10.6.0 software.

## MTT viability assays

For dose-response to drugs, cell viability was assessed using MTT viability assays. Cells were plated in triplicates at 15,000 cells/well in 96-well plates and left for 2 h to adhere prior to drug addition. Cells were left to grow for 72 h, and MTT diluted in PBS was added at 5 μg/ml. After 4 h incubation, medium was removed and replaced by 150 μl DMSO, and 570 nm absorbance was performed on a BMG Labtech ClarioStar plate reader. Triplicates were averaged and normalized by untreated controls.

## Trypan blue proliferation and viability assays

For proliferation and viability assays, cells were plates at 100,000 cells/well in 6-well plates. Cells were then detached, resuspended in 500 μl medium, and live/dead cells were counted using a Beckman Coulter Vi-Cell cell counter.

## RNA sequencing

Following centrosome amplification with doxycycline for PLKOE versus DMSO for PLK4Ctl, total RNA was extracted with RNeasy Mini kit (Qiagen #74104) following manufacturer's instructions. RNA integrity and quality were checked with Agilent RNA 6000 Nano Kit (Agilent, #5067–1511) and corresponding devices. Samples were processed at Institut Curie NGS platform from cDNA synthesis, amplification, quality assessment, and sequencing. Novaseq 6000 system (Illumina) was used for sequencing (read length of 100 bp, paired end). All the bioinformatic analysis were done by Genosplice (http://www.genosplice.com) including quality control of sequences generated, read mapping, and gene differential analysis (R software, Deseq2). Biological interpretation of the identified genes was done using GSEA tool for pathway enrichment analysis between distinct conditions.

## Single-cell whole genome sequencing

Cells were treated with DMSO (1/10,000), Doxycycline (1 μg/ml), or AZ3146 (1 μm) for 72 h. Cells were then frozen in freezing medium (10% DMSO, 40% FBS in DMEM-F12).

## Nuclei preparation and sorting

Cells were thawed, and single-cell sequencing was performed on cell nuclei isolated from cell lysis, leaving the nucleus intact. Thawed cells were prepared by resuspending in PBS + 1% BSA, washing, and pelleting. To generate nuclei, cells were resuspended and incubated (15 min on ice in dark environment) in cell lysis buffer (100 mM Tris-HCl (pH 7.4), 154 mM NaCl, 1 mM CaCl2, 500 μm MgCl2, 0.2% BSA, 0.1% NP-40, 10 μg/ml Hoechst 33358, 2 μg/ml propidium iodide in ultra-pure water). Resulting cell nuclei were gated for G1 phase (as determined by Hoechst and propidium iodide staining) and sorted into wells of 96 wells plates on a MoFlo Astrios cell sorter (Beckman Coulter), depositing 1 cell per well, and 96 wells plates containing nuclei and freezing buffer were stored at −80°C until further processing. Automated library preparation was then performed as previously described [54].

## AneuFinder analysis

Sequencing was performed using a NextSeq 2000 machine (Illumina; up to 120 cycles; single end or up to 113 and 7 cycles; paired end). The generated data were subsequently demultiplexed using sample-specific barcodes and changed into fastq files using bcl2fastq (Illumina; version 1.8.4). Reads were afterwards aligned to the human reference genome (GRCh38/hg38) using Bowtie2 (version 2.2.4; [55]). Duplicate reads were marked with BamUtil (version 1.0.3; [56]).

The aligned read data (bam files) were analyzed with a copy number calling algorithm called AneuFinder (version 1.14.0; [57]) using an euploid reference [58]. Following GC correction (R package: BSgenome.Hsapiens.UCSC.hg38_1.4.1; The Bioconductor Dev Team 2015) and blacklisting of artefact-prone regions (extreme low or high coverage in control samples), libraries were analyzed using the dnacopy and edivisive copy number calling algorithms with variable width bins (average binsize = 1 Mb; step size = 500 kb) and breakpoint refinement (refine.breakpoints = TRUE). Results were afterwards curated by requiring a minimum

concordance of 90% between the results of the 2 algorithms. Libraries with on average less than 10 reads per bin and per chromosome copy (approximately 55,000 reads for a diploid genome) were discarded.

### Aneuploidy score

The aneuploidy score of each bin was calculated as the absolute difference between the observed copy number and the expected copy number when euploid. The score for each library was calculated as the weighted average of all the bins (size of the bin as weight) and the sample scores were calculated as the average of the scores of all libraries.

### Heterogeneity score

The heterogeneity score of each bin was calculated as the proportion of pairwise comparisons (cell 1 versus cell 2, cell 1 versus cell 3, etc.) that showed a difference in copy number (e.g., cell 1: 2-somy and cell 2: 3-somy). The heterogeneity score of each sample was calculated as the weighted average of all the bin scores (size of the bin as weight).

### Centrosome numbers in tumors

For each sample, 10 randomly chosen fields were considered. Using ImageJ software, we visually counted the number of nuclei and the number of centrosomes in a blind manner without taking into account tumor identity. The centrosome to nuclei ratio (CNR) was obtained by dividing the total number of centrosomes by the total number of nuclei in each field.

### Proliferation and mitotic index

For Ki67 proliferation assessment, we performed immunochemistry assays using mouse anti-human ki67 antibody (M7240, DAKO, 1/200 at pH 9) in a series of paraffin-embedded tissue blocks of HGSOC. Sections of 3 μm were cut using a microtome from the paraffin-embedded tissue blocks of normal tissue and invasive lesions. Tissue sections were deparaffinized and rehydrated through a series of xylene and ethanol washes. Briefly, the key steps included: (i) antigen retrieval with ER2 pH9, (Leica: AR9640); (ii) blocking of endogenous peroxidase activity with Bond polymer refine detection kit (Leica: DS9800); (iii) incubation with primary antibodies against the targeted antigen; and (iv) immunodetection with Revelation and counter staining Bond polymer refine detection kit (Leica: DS9800). Immunostaining was performed using a Leica Bond RX automated immunostaining device. We performed an immunohisto-chemical score (frequency x intensity) through analysis of 10 high-power fields (HPFs, x 400). All quantifications were performed by 2 pathologists with blinding of patient status.

For mitotic index, paraffin-embedded tissue sections of tumors were stained with hematoxylin and eosin. The mitotic count was determined by the number of mitotic figures found in 10 consecutive HPFs, in the most mitotically active part of the tumor (entire section). Only identifiable mitotic figures were counted. Hyperchromatic, karyorrhectic, or apoptotic nuclei were excluded.

### Statistical analysis

Statistical analysis was performed using GraphPad Prism. The tests used are specified in the figure legends. The numbers of cells analyzed and the number of replicates are reported either on the figure or in the respective figure legends.

## Supporting information

**S1 Fig.** **(A)** Representative images of OVCAR8 cells stained with DAPI (gray) and antibodies against CEP192 (Cyan) and Pericentrin (Magenta). **(B)** Bar graphs showing the averages and SEM of the percentage of cells with the indicated number of centrosomes (CEP192 dots colocalizing with Pericentrin). Three independent experiments, statistical test: Fisher's exact test comparing the number of cells with more than 2 centrosomes. **(C)** Dose-response of PLK4Ctl and PLK4OE cells to Paclitaxel (left) and Carboplatin (right), normalized to their respective control conditions, obtained from MTT viability assays. Mean and SEM of 2 independent experiments each obtained from averaging 3 technical replicates. **(D)** Combination matrixes for Carboplatin and Paclitaxel combined treatment, representing percentage of viability inhibition compared to control cells. Chosen working concentrations are highlighted in red. **(E)** Scatter dot plots of Interphase length (top) and First mitosis length (bottom), with Median and interquartile range. Data from 2 independent experiments are pooled for Combined treatment and Carboplatin treatment, data from the 4 corresponding control experiments are pooled for Untreated. For interphase length a minimum of 26 cells was analyzed, and for mitosis length a minimum of 133 cells was analyzed. Statistical tests: Kruskal–Wallis with Dunn's multiple comparisons tests. **(F)** Single-cell profiles of PLK4Ctl (left) and PLK4OE (right) Untreated cells. Color coding of mitosis and fates refers to categories defined in Fig 1A. **(G)** Averages and SEM of the percentages of mitotic phenotypes (legends in Fig 1A and 1F). Two independent experiments, statistical test: Fisher's exact test on the number of Slight Mis-segregation events. **(H)** Percentage of multipolar divisions observed in presence or absence of 5 nM Paclitaxel. Two independent experiments, statistical test: Fisher's exact test on the number of multipolar divisions. **(I)** Representative images of OVCAR8 cells in prometaphase/metaphase stained with DAPI (blue), and antibodies against α-Tubulin (green), CEP192 (magenta), and Centrin3 (cyan) Numbered insets to show centrosomes are presented below the spindle image. Scale bars are 10 μm for spindle images, and 1 μm for centrosome insets. Spindles of the "undefined" category are only seen in cells treated with paclitaxel. **(J)** Averages and SEM of the percentages of mitotic phenotypes identified in panel I. Two independent experiments, statistical test: Fisher's exact test on the total number of bipolar and pseudo-bipolar spindles. Data for S1 Fig can be found in S5 Data.
(TIFF)

**S2 Fig.** **(A)** Representative images of OVCAR8 cells in anaphase stained with DAPI (gray), and antibodies against α-Tubulin (green), CEP192 (magenta) and Centrin3 (cyan). Numbered insets to show centrosomes and spindle poles are presented below the spindle image. Scale bars are 10 μm for spindle images and 1 μm for centrosome insets. **(B)** Averages and SEM of the percentages of spindle types shown in panel C. Two independent experiments, statistical test: Fisher's exact test on the number of bipolar spindles. **(C)** Averages and SEM of the percentages of Carboplatin treated PLK4Ctl cells presenting additional MTOCs with the indicated characteristics. **(D)** Representative image of a Carboplatin treated PLK4OE OVCAR8 cell in anaphase stained with DAPI (gray), and antibodies against α-Tubulin (green), CEP192 (magenta) and CREST (cyan). Numbered insets to show chromosomes with (1) or without (2) centromeres. Scale bars are 10 μm for spindle image and 1 μm for chromosome insets. **(E)** Averages and SEM of the percentages of mitotic figures depending on the presence of acentric chromosomes. Two independent experiments, statistical test: Fisher's exact test on the total number of bipolar and pseudo-bipolar spindles. **(F)** Averages and SEM of the percentages of mitotic behaviours. Two independent experiments, statistical test: Fisher's exact test on the number of High mis-segregation events. **(G)** Averages and SEM of the percentages of the indicated mis-segregation events within bipolar divisions. Two independent experiments,

statistical test: ANOVA with Sidak's multiple comparison test. (H) Time lapse stills of iOV-CAR8 cells expressing GFP-Tubulin (green) and H2B-RFP (red) to visualize the mitotic spindle and chromosomes treated with DOX and carboplatin. Time is shown in minutes. White arrows point to centrosomes that do not show a clustering behavior, while yellow arrows point to clustered centrosomes. In the top row, a cell enters mitosis with at least 5 centrosomes, it goes through a multipolarity status before achieving bipolarity and dividing in a bipolar manner. The second row shows a cell with at least 4 centrosomes that cluster to form a bipolar spindle. The white arrow at time point 70.00 min shows a centrosome that emerges from the bipolar spindle but does not result in multipolarity. The third row shows a cell with severe chromosome mis-alignment and mis-segregation defects, where the spindle goes through cycles of collapsing and bipolarization. This cell takes over 10 h to divide. The fourth row shows a cell with at least 6 centrosomes at mitotic entry. These are clustered in 3 main poles and the cell divides in a multipolar manner. The fifth cell shows a cell dying during mitosis. Note that mitotic time is not extended as in other examples. (I) Averages and SEM of the percentages of mitotic behaviors. Color scheme is defined in S2F Fig. Two independent experiments, statistical test: Fisher's exact test on the number of High mis-segregation events. Data for S2 Fig can be found in S6 Data.
(TIFF)

**S3 Fig.** (A, B) Single-cell profiles of FUCCI PLK4Ctl (A) and PLK4OE (B) untreated cells. See panel C for color-coded legends of cell cycle phase and cell fate. (C) Bar graphs showing the averages and SEM of the percentages of cell death events occurring in the indicated cell-cycle phases. Two independent experiments, statistical test: Fisher's exact test on number of death events occurring in S/G2. (D, E) Scatter dot plot graph of S/G2 (D) and G1 (E) phase lengths in the second generation, with median and interquartile range. Two independent experiments with a minimum of 18 times analyzed per category. Statistical tests: Kruskal–Wallis with Dunn's multiple comparisons tests. (F) Scatter dot plot graph of G1 phase length in the second generation, with median and interquartile range, depending on the fate of the cells in the second generation. Two independent experiments with a minimum of 70 times analyzed per category. Statistical tests: Kruskal–Wallis with Dunn's multiple comparisons tests. (G) Representative images of western blot analysis of phosphorylated Chk1 and p53. (H) Graph showing the average and SEM of phosphorylated protein relative to total protein levels, normalized to the levels detected in PLK4OE cells treated with Carboplatin for 24 h, from 2 independent experiments. (I) Representative images of cells stained with DAPI an antibodies against FANCD2 (cyan), 53BP1 (gray), and CEP192 (magenta). Grayscale images of RAD51 and γH2AX are shown. (J) Dot-plot representing integrated nuclear γH2AX fluorescence intensity per cell or numbers of Rad51, FANCD2, or 53BP1 foci per cells. Average and SEM of the averages obtained from 3 independent experiments, each quantifying a minimum of 94 cells per condition. Values are normalized to the average of untreated PLK4Ctl cells at 24 h. Statistical test: ANOVA with Sidak's multiple comparison tests. Data for S3 Fig can be found in S7 Data.
(TIFF)

**S4 Fig.** **(A)** Representative images of western blot analysis of p21 and PUMA. **(B)** Graph showing the average and SEM of protein levels from 4 (p53 and p21) or 3 (PUMA) independent experiments, normalized to levels measured in untreated PLK4Ctl cells at 48 h. **(C and G)** Representative images of western blot analysis of indicated shRNA cell lines. **(D and H)** Bar graphs showing the average and SEM of the percentage of cells in specified Annexin V-APC/PI gates analyzed by flow cytometry. (D) Six replicates obtained from 4 independent experiments, with a minimum of 10,000 cells analyzed per condition and replicate. (H) Four

replicates obtained from 2 independent experiments, with a minimum of 10,000 cells analyzed per condition and replicate. Statistical test: comparison of the percentage of Annexin V positive cells, using ANOVA with Sidak's multiple comparison test. Representative cytometry profiles can be found in the S1 Appendix. **(E)** Representative images of western blot analysis of Caspase2 and MDM2 cleavage. **(F)** Average and SEM of protein levels from 3 (p19 Caspase2) or 2 (p55 MDM2) independent experiments, normalized to levels measured in untreated PLK4OE cells at 24 h. Data for S4 Fig can be found in S8 Data.
(TIFF)

**S5 Fig. (A, B)** Dose-response of PLK4Ctl and PLK4OE cells to A1210477 **(A)** and Venetoclax **(B)**, normalized to their respective untreated conditions, obtained from MTT viability assays. Mean and SEM of 2 independent experiments each obtained from averaging 3 technical replicates. **(C)** Bar graphs showing the average and SEM of the percentage of cells with the indicated number of centrosomes (CEP192 dots colocalizing with Pericentrin). Two independent experiments, statistical test: Fisher's exact test comparing the number of cells with more than 2 centrosomes. Numbers on the top of each graph represent the number of cells analyzed per condition. **(D)** Bar graphs showing the average and SEM of the percentage of cells with the indicated number of centrosomes (CEP192 dots colocalizing with Pericentrin). Two independent experiments, statistical test: Fisher's exact test comparing the number of cells with more than 2 centrosomes. Numbers on the top of each graph represent the number of cells analyzed per condition. **(E)** Representative images of COV504 and SKOV3 cells stained with DAPI (gray) and antibodies against CEP192 (cyan) and Pericentrin (magenta). **(F)** Bar graphs showing the average and SEM of the percentages of cells with the indicated number of centrosomes (CEP192 dots colocalizing with Pericentrin). Three independent experiments, statistical tests: Fisher's exact tests comparing the number of cells with more than 2 centrosomes. Numbers on the top of each graph represent the number of cells analyzed per condition. **(G)** Dose-response of PLK4Ctl and PLK4OE OVCAR8 (top), COV504 (middle), and SKOV3 (bottom) to Navitoclax, normalized to their respective untreated conditions, obtained from MTT viability assays. Mean and SEM of 2 to 3 independent experiments each obtained from averaging 3 technical replicates. **(H)** Table summarizing the IC50s and drug concentrations used in combinations, determined via MTT dose-response viability assays, for PLK4Ctl OVCAR8, COV504, and SKOV3. **(I)** Viability (% of Trypan Blue negative cells) counted for PLK4Ctl and PLK4OE OVCAR8 (top), COV504 (middle), and SKOV3 (bottom), in response to indicated chemotherapies using concentrations indicated in panel G for 72 h. Average and SEM of the number of independent experiments indicated, each obtained from averaging 3 technical replicates. Statistical test: two-way ANOVA with Sidak's multiple comparison test. **(J)** Percentage of multipolar divisions observed in COV504 in response to 5 nM Paclitaxel exposure. **(K)** Viability (% of Trypan Blue negative cells) counted for PLK4Ctl and PLK4OE OVCAR8 (top), COV504 (middle), and SKOV3 (bottom), in response to Olaparib using concentrations indicated in panel G for 72 h. Average and SEM of 4 independent experiments, each obtained from averaging 3 technical replicates. Statistical test: two-way ANOVA with Sidak's multiple comparison test. Data for S5 Fig can be found in S9 Data.
(TIFF)

**S6 Fig. (A)** Genome-wide copy-number plots for G1 OVCAR8 cells. Each row represents a cell. Indicated aneuploidy and heterogeneity scores are calculated as described in Material and methods. **(B)** Bar graphs showing the average and SEM of the percentage of mitotic phenotypes as defined in Fig 1A. Two independent experiments, statistical test: comparison of the percentage of cells with no mis-segregation using ANOVA with Dunnett's multiple comparison test. Numbers on the top of each graph represent the number of cells analyzed per

condition. **(C)** Scatter dot plot graph of mitosis length with median and interquartile range. At least 96 mitosis analyzed from 2 independent experiments, statistical test: Kruskall–Wallis with Dunn's multiple comparison test. **(D)** Bar graphs showing the average and SEM of the percentage of cells in specified Annexin V-APC/PI gates analyzed by flow cytometry. Three replicates obtained from 2 independent experiments, with a minimum of 20,000 cells analyzed per condition and replicate. Statistical test: comparison of the percentage of Annexin V positive cells, using ANOVA with Sidak's multiple comparison test. Representative cytometry profiles can be found in the S1_Appendix. **(E)** GSEA Hallmarks with |normalized enrichment score >1,5 and $p$-value <0,05, from differential RNA expression analysis of PLK4OE cells compared to PLK4Ctl. **(F)** Western blot analysis of STING phosphorylation after 72 h of indicated drug treatments with quantification of pSTING relative to total STING. Average and SEM of 2 independent experiments, normalized to untreated PLK4Ctl. HT-DNA transfection was performed 24 h before cell collection. **(G)** Western blot analysis of indicated bulk Lenti-CRISPR cell lines. **(H)** Bar graphs showing the average and SEM of the percentage of cells in specified Annexin V-APC/PI gates analyzed by flow cytometry. Four replicates obtained from 2 independent experiments with a minimum of 15,000 cells analyzed per condition and replicate. Statistical test: comparison of the percentage of Annexin V positive cells using ANOVA with Sidak's multiple comparison test. Representative cytometry profiles can be found in the S1_Appendix Numbers on the top of each graph represent the number of cells analyzed per condition. Data for S6 Fig can be found in S10 Data.
(TIFF)

**S7 Fig.** **(A)** Average and SEM of the CNR established in 10 fields per tumor or healthy tissue sample. Red dotted line indicates CNR = 1,45, the cut-off between High-CNR and Low-CNR patients. **(B, C)** Distribution of the Mitotic Index (B) and of the percentage of Ki67 positive cells (C) as a function of CNR. Statistical test: Spearman correlation. **(D)** Average and SEM of CNR per patient classified depending on FIGO stage. **(E, F)** Kaplan–Meier curves for overall survival (B) and relapse-free time after the first line of chemotherapy (C) according to CNR status. Data for S7 Fig can be found in S11 Data.
(TIFF)

**S1 Movie. Representative movies of cells undergoing death in interphase (left) or death in mitosis (right).** Brightfield and H2B-RFP channels are shown in gray and yellow, respectively.
(MOV)

**S2 Movie. Representative movies of untreated PLK4Ctl (left) and PLK4OE (right) cells.** Brightfield and H2B-RFP channels are shown in gray and yellow, respectively.
(MOV)

**S3 Movie. Representative movies of combined chemotherapy treated PLK4Ctl (Left) and PLK4OE (Right) cells.** Brightfield and H2B-RFP channels are shown in gray and yellow, respectively.
(MOV)

**S4 Movie. Representative movies of the indicated mitotic behaviors.** H2B-RFP channel is shown in gray, except for Failed Cytokinesis and Slippage where brightfield and H2B-RFP channels are shown in gray and yellow, respectively.
(MOV)

**S5 Movie. Representative movies of carboplatin treated PLK4Ctl (left) and PLK4OE (right) cells.** Brightfield and H2B-RFP channels are shown in gray and yellow, respectively.
(MOV)

**S1 Table. Tumor sample description used in this study and obtained through the Institut Curie hospital.**
(PDF)

**S2 Table. List of plasmids used in this study.**
(PDF)

**S3 Table. List of chemicals used in this study.**
(PDF)

**S4 Table. List of antibodies used in this study.**
(PDF)

**S1 Appendix. Representative cytometry profiles.**
(PDF)

**S1 Raw Images. Original western blot gels.**
(PDF)

**S1 Data. Excel spreadsheets containing, in separate sheets for each figure, the underlying and individual numerical data used for Fig 1B–1F.**
(XLSX)

**S2 Data. Excel spreadsheets containing, in separate sheets for each figure, the underlying and individual numerical data used for Fig 2A–2K.**
(XLSX)

**S3 Data. Excel spreadsheets containing, in separate sheets for each figure, the underlying and individual numerical data used for Fig 3A–3C and 3E and 3F.**
(XLSX)

**S4 Data. Excel spreadsheets containing, in separate sheets for each figure, the underlying and individual numerical data used for Fig 4A, 4C, 4E, 4F, 4H and 4I.**
(XLSX)

**S5 Data. Excel spreadsheets containing, in separate sheets for each figure, the underlying and individual numerical data used for S1B–S1H and S1J Fig.**
(XLSX)

**S6 Data. Excel spreadsheets containing, in separate sheets for each figure, the underlying and individual numerical data used for S2B, S2C, S2E–S2G, and S2I Fig.**
(XLSX)

**S7 Data. Excel spreadsheets containing, in separate sheets for each figure, the underlying and individual numerical data used for S3A–S3H and S3J Fig.**
(XLSX)

**S8 Data. Excel spreadsheets containing, in separate sheets for each figure, the underlying and individual numerical data used for S4B and S4D Fig.**
(XLSX)

**S9 Data. Excel spreadsheets containing, in separate sheets for each figure, the underlying and individual numerical data used for S5A–S5D and S5F–S5J Fig.**
(XLSX)

**S10 Data. Excel spreadsheets containing, in separate sheets for each figure, the underlying and individual numerical data used for S6B–S6D and S6E–S6H Fig.**
(XLSX)

**S11 Data. Excel spreadsheets containing, in separate sheets for each figure, the underlying and individual numerical data used for S7A–S7F Fig.**
(XLSX)

## Acknowledgments

We are grateful to the patients who consented to participate in this research and to the medical teams involved in their care. We thank Andrew Holland for the kind gift of the inducible PLK4 overexpression plasmid. We thank the ICGEX platform at Institut Curie (IC) headed by Sylvain Baulade for bulk RNA sequencing and Genosplice (http://www.genosplice.com) and P. Delagrange and W. Zeitouni for performing the analysis. We thank the Tissue Imaging (PICT-IBiSA) and Nikon Imaging Centre at IC, member of the French National Research Infrastructure France-BioImaging (ANR10-INBS-04) and the IC cytometry platform. We thank the Genomics Platform of the translational research department at IC for cell line authentication. We thank Nicolas Manel and Sebastian Montealegre for discussions and the gift of reagents used to characterize cGAS/STING signaling. We thank Stephen Taylor and Laura Attardi for discussions about this project. We also thank all the members of the Basto team for discussions and comments on the manuscript.

## Author Contributions

**Conceptualization:** Frances Edwards, Oumou Goundiam, Renata Basto.

**Data curation:** Odette Mariani.

**Formal analysis:** Frances Edwards, Andrea E. Tijhuis, Rene Wardenaar, Oumou Goundiam.

**Funding acquisition:** Frances Edwards, Renata Basto.

**Investigation:** Frances Edwards, Giulia Fantozzi, Anthony Y. Simon, Jean-Philippe Morretton, Aurelie Herbette, Andrea E. Tijhuis, Rene Wardenaar, Stacy Foulane, Simon Gemble, Oumou Goundiam.

**Methodology:** Frances Edwards.

**Project administration:** Renata Basto.

**Resources:** Oumou Goundiam, Renata Basto.

**Software:** Frances Edwards.

**Supervision:** Frances Edwards, Diana C.J. Spierings, Floris Foijer, Anne Vincent-Salomon, Sergio Roman-Roman, Xavier Sastre-Garau, Oumou Goundiam, Renata Basto.

**Visualization:** Frances Edwards, Giulia Fantozzi, Anthony Y. Simon.

**Writing – original draft:** Frances Edwards.

**Writing – review & editing:** Frances Edwards, Renata Basto.

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
