## [Editor Report · Decision Letter 0]

30 Nov 2023

Dear Renata,

Thank you for submitting your manuscript from Review Commons entitled "Centrosome amplification primes for apoptosis besides promoting multipolar divisions, and potentiates the response to chemotherapy in ovarian cancer" for consideration as a Research Article by PLOS Biology. Please accept my apologies for the delay in getting back to you as we consulted with an academic editor about your submission.

Your manuscript has now been evaluated by the PLOS Biology editorial staff, as well as by an academic editor with relevant expertise, and I am writing to let you know that we would like to invite you to submit a revised version in response to the reports at Review Commons.

However, before we can invite a revision, we need you to complete your submission by providing the metadata that is required for full assessment. To this end, please login to Editorial Manager where you will find the paper in the 'Submissions Needing Revisions' folder on your homepage. Please click 'Revise Submission' from the Action Links and complete all additional questions in the submission questionnaire.

To provide the metadata for your submission, please Login to Editorial Manager (https://www.editorialmanager.com/pbiology) within two working days, i.e. by Dec 02 2023 11:59PM.

Best wishes,

Richard

Richard Hodge, PhD

rhodge@plos.org

PLOS

---

## [Editor Report · Decision Letter 1]

11 Dec 2023

Dear Renata,

Thank you very much for submitting your manuscript "Centrosome amplification primes for apoptosis besides promoting multipolar divisions, and potentiates the response to chemotherapy in ovarian cancer" for consideration as a Research Article at PLOS Biology. As you know, your manuscript and plan of revision have been evaluated by the PLOS Biology editors and by an Academic Editor with relevant expertise.

Based on your responses to the reviews from Reviews Commons, we would welcome re-submission of a revised version that takes into account the reviewers' comments according to your revision plan. After discussions with the Academic Editor, we would also like to consider your manuscript as a Short Report at the journal given the comments raised by the reviewers regarding the depth of mechanistic insights provided for the observations (https://journals.plos.org/plosbiology/s/what-we-publish#loc-short-reports). Upon resubmission, we would be grateful if you could please tick ‘Short Report’ as the article type in the drop-down menu. In addition, our Short Report format has a maximum of 4 main figures, so we ask that you please reduce the number of main figures by either combining some of the figures or by moving to the supplementary. 

Given the extent of revision needed, we cannot make any decision about publication until we have seen the revised manuscript and your response to the reviewers' comments at Review Commons. Your revised manuscript is also likely to be sent for further evaluation by the original reviewers.

We expect to receive your revised manuscript within 3 months. Please email us (plosbiology@plos.org) if you have any questions or concerns, or would like to request an extension. At this stage, your manuscript remains formally under active consideration at our journal; please notify us by email if you do not intend to submit a revision so that we may withdraw it.

**IMPORTANT - SUBMITTING YOUR REVISION**

*Re-submission Checklist*

*Published Peer Review*

*PLOS Data Policy*

*Blot and Gel Data Policy*

Best wishes,

Richard

Richard Hodge, PhD

rhodge@plos.org

REVIEWS:

Reviewer #1: In this manuscript, Edwards et al analyze OVCAR8 cells with dox inducible expression of Plk4. Doxycycline treatment induces centrosome amplification in ~80% of cells. 72 hour timelapse analysis of cells with fluorescent chromosomes revealed that cell death after carboplatin+paclitaxel was more common in Plk4OE than Plk4Ctl cells. Cell death was most common after high chromosome mis-segregation/multipolar division, which resulted in death of ~80% of the daughter cells. However, death was also elevated in cells with no or slight mis-segregation when comparing Plk4OE to PlkrCtl (40% vs 12%), suggesting an additional sensitization effect. Plk4OE also increased cell death in carboplatin alone, most notably after high mis-segregation but also to a lesser extent in cells with no or slight mis-segregation. 17% of Plk4OE cells exposed to carboplatin in G1 died in S/G2 vs 6% of Plk4Ctl. This difference did not appear to be due to DNA damage response or PIDDosome activity. Carboplatin caused caspase 3 cleavage and cytochrome C release to a greater extent in Plk4OE than Pl4Ctl cells, suggesting MOMP priming. Plk4OE sensitizes OVCAR8 cells to the BCL-XL inhibitor WEHI-539, and Plk4OE sensitized COV504 but not SKOV3 cells to the less specific inhibitor Navitoclax. In 88 patients with high grade serous ovarian carcinoma, a high (>1.45) centrosome-to-nucleus ratio was associated with increased relapse-free and overall survival. The authors conclude that centrosome amplification primes ovarian cancer cells to chemotherapy independent of mitotic defects. 

**Major comments** 

1. In its current form, the title suggests that the major role of centrosome amplification in sensitizing to chemotherapy is independent of multipolar divisions. Based on Figure 1, this is misleading. Figure 1D shows that in centrosome amplified cells treated with combination chemotherapy, the most common cause of death is high mis-segregation on multipolar spindles. Modifying the title to "Centrosome amplification favors the response to chemotherapy in ovarian cancer by priming for apoptosis in addition to promoting multipolar division" would more accurately reflect the data. 

2. In a previous technical tour de force (Morretton et al, EMBO Mol Med 2022), the Basto lab quantified centriole numbers in the ovarian patient cancer samples analyzed here, and found that the percentage of cells with centrosome amplification in a given ovarian tumor is quite small, only reaching a maximum of 3.2%. It is critical background information to cite that quantification here. This information also begs the question of whether introducing this low rate of centrosome amplification is sufficient to cause a more global apoptotic priming in the sample, as suggested. 

3. The conclusion that centrosome amplification primes to apoptosis irrespective of mitotic defects is largely based on low resolution timelapse analysis (20x magnification, 10 minute imaging intervals, no tubulin). Imaging at this resolution is likely to miss mitotic defects, reducing the confidence with which this conclusion can be drawn. 

4. Data from timelapse analysis of DNA content in Fig. 2 are used to conclude that Plk4OE cells are more sensitive to carboplatin due to mitotic defects that occurred without multipolar spindles. However, it is premature to conclude that multipolar spindles were not involved in DNA mis-segregation without visualizing the spindles themselves. While DNA positioning can be used as a proxy for spindle morphology, as performed here, it only reliably detects multipolar spindles when all poles are relatively equal in size and the multipolar spindle is maintained throughout mitosis. However, the poles in multipolar spindles often differ in size and ability to recruit DNA. Additionally, they often cluster over time, which can preclude their identification when only visualizing DNA, especially at 20x magnification. Compelling evidence that high mis-segregation is occurring without multipolar spindles would require visualizing the spindles and also demonstrating the cause of the increased chromosome missegregation. (Are acentric fragments being mis-segregated as lagging chromosomes?) 

5. The images in Fig. 3D and 4D are not of sufficient resolution to support the central conclusion that centrosome amplification primes cells for MOMP. This conclusion is further weakened by the facts that 1) Plk4OE was the only source of centrosome amplification tested and 2) Plk4OE was reported to prime for MOMP in only 2 of 3 cell lines. Potential explanations for the lack of priming in SKOV3 cells should be discussed. Additionally, the sensitization in Fig. S4H-J appears quite modest. (These data are also difficult to see, perhaps because the Plk4Ctl -/+ chemo conditions are overlapping.) 

6. Line 191 points out that more Plk4OE cells that were in G1 at the beginning of carboplatin died than Plk4Ctl cells. However, in Fig. 2H-I, it looks like longer G1 durations in the presence of carboplatin led to increased cell death and that the Plk4OE cells happened to spend more time in G1 at the beginning of carboplatin treatment than Plk4Ctl cells did. Is this the case? Quantification of the average time spent in G1 for each group would be helpful. 

**Minor comments** 

7. The authors cite Fig. 1B when drawing the conclusion that "combined chemotherapy induced a stronger reduction of viable cells produced per lineage in PLK4OE compared to PLK4Ctl". But Fig. 1B shows that combination chemotherapy produced a similar decrease in viable cells per lineage +/- Plk4OE. If anything, the Plk4OE+ cells showed slightly less sensitivity because they proliferated more poorly in the absence of chemotherapy. This is also true for carboplatin sensitivity in Fig. 2D (line 156). 

8. Line 202 concludes that Fig. S2H-I shows that Plk4OE doesn't affect recruitment of DNA damage repair factors. The dotted outlines around the nuclei in Fig. S2H-1 make it very difficult to see, but it appears that gH2AX, FancD2, and 53BP1 signals are lower in Plk4OE cells. 

9. The images for "dies in interphase" and "dies in mitosis" in Fig. 1B are suboptimal. Alternative images would be beneficial. 

10. It would be helpful to discuss the clinical relevance of WEHI-539 and Navitoclax. 

11. The discussion states that "mitotic drugs that limit centrosome clustering have had limited success in the clinic". I am not aware of any drugs that limit centrosome clustering that are suitable for in vivo use and the citation provided does not mention centrosome clustering. 

12. The dark purple and black are very difficult to discriminate (Figure 1,2 and S1), as are the light green and light turquoise (Fig. 4A,S4A-B, S4F, S4H). 

13. Line 246 claims that Fig. S3B shows that p21 and PUMA mildy increase upon carboplatin exposure, but it isn't clear that these increase in a biologically or statistically significant manner. 

14. The green used to indicate S/G2 in Fig. S2A-B is different in Plk4 Ctrl vs Plk4OE cells. 

15. I do not believe that carboplatin + paclitaxel is standard of care treatment for breast or lung cancer, as stated on line 48-49. 

16. This study advances, but does not complete our understanding of centrosome amplification in breast cancer, as stated on line 75. 

17. Line 297 describes Navitoclax as an "inhibitor of BCL2, BCL-XL and BCL2". (ie BCL2 is listed twice). 

18. It's not clear why line 120, which refers to effects of combined chemotherapy, cites Fig. S1G-I, which apparently show data from untreated (even without dox?) Plk4Ctrl and Plk4OE cells. 

19. In Fig. S6A, how can the mitotic index be 200%? 

The importance of centrosome amplification in cancer has long been debated. The possible effects of extra centrosomes on multipolar divisions are well known. An independent apoptosis-priming effect of additional centrosomes is novel and of interest. However, in their previous manuscript (Morretton et al, EMBO Mol Med 2022), the Basto lab showed that centrosome amplification only occurs in a maximum of 3.2% of cells in a given ovarian cancer. Given the large discrepancy between the rate of centrosome amplification in the models here and in ovarian cancers ({greater than or equal to}80% vs {less than or equal to}3%), it is unclear whether the mechanism of apoptosis priming reported here is at play in a clinical setting. It is unclear whether the low rate of centrosome amplification observed in cancers can predispose response to a particular inhibitor, as suggested, particularly when centrosome amplification in {greater than or equal to}80% of cells 1) only induced apoptosis priming in 2 of 3 cell lines (Fig. S4F) and 2) induced relatively modest drug sensitivity (Fig. S4J). If it were shown in an additional experiment that induction of centrosome amplification in a small minority of cells, as occurs in patient tumors, increases MOMP priming and drug response, this would substantially increase the significance of the study.

Reviewer #2: **Summary** The authors' findings suggest that induction of centrosome amplification synergises with and potentiates the cytotoxic effects of standard chemo in epithelial ovarian cancer cell lines via a mechanism involving mitochondrial membrane priming and Cyt C release. CA has differential apoptotic priming effects depending on the cell line context. The authors use single cell analysis to characterise the range of mitotic defects through to cell fate following PLK4 OE and the combination treatments. The studies are extended to an ovarian cancer patient cohort where elevated centrosome numbers are associated with better OS and RFS. These findings have the potential to improve future patient stratification and treatment in EOC in addition to prognosis of treatment response. 

**Major comments** 

1. The authors state that (Line 133) "the increased multipolarity we observe in presence of the combined chemotherapy is caused by the effect of Paclitaxel on the capacity of cells to cluster centrosomes." Could the authors to back up this claim by reanalysing the imaging data to look for clustering as a survival mechanism versus inhibition of clustering in Paclitaxel-treated cells? Or indeed test any of the range of available clustering inhibitors directly on PLK4OE and thus prove the contribution of clustering to survival? 

2. Fig 3: Results line 229-236 refer to quantification of fragmented nuclei which the authors interpret as poised for apoptosis. Micronuclei are also quantified- do the authors interpret this phenotype as advanced apoptosis? There is no mention of apoptotic bodies in the analysis. I would ask the authors to provide a bank of representative images with explanations to illustrate their interpretation of the range of morphologies - differences between nuclear fragmentation, versus micronuclei versus DNA contained in apoptotic bodies. 

3. Fig 6: While the authors have already acknowledged this as a weakness of the study, can the patient data really be compared to cell line data on CA because inclusion of CNRs between 1.4 and 2 as "high CNR" is questionable given that this ratio represents a completely normal centrosome complement? Are the authors confident enough in the imaging technique that all centrosomes are being detected? Can the authors justify the inclusion of the 1.4-2 CNR tumours by breaking down individual patient data on response to various treatments? Have the authors tried to analyse the cohort for OS and RFS using only those 9% of tumours exhibiting CA? What does the analyses of Fig 6 and S6 look like with a CNR cut-off of 2 instead of 1.45? Does the re-analysis show a better correlation between CNR and FIGO stage? 

4. Although this patient cohort is described in a previous publication, authors should include a cohort description in a table within supplemental for this manuscript: age range of patients, number of patients in each stage, size of tumours, and most relevant to this study, treatment regimens- adjuvant versus neoadjuvant, surgery vs no surgery? How is the cohort selectedsequentially selected? inclusion/exclusion criteria? Statement in abstract "we show that high centrosome numbers associate with improved chemo responses" is too specific as we have no information on the treatment regimens received by the patients (neo or adjuvant chemo versus surgical/radiological interventions?). Maybe treatment response would be more appropriate? Were there any cases of Pathological complete (or even near complete) response in this cohort and if so, what was the CNR in those cases? 

5. The experimental PLK4 overexpression system is an accepted and clean method to induce CA in vitro. Could the authors comment in the discussion on how they envision CA being induced as a sensitizing agent in the clinical setting to support the translational aspects of their work? 

**Minor comments** 

The manuscript is well written and all data clearly and thoroughly presented. 

Just some minor points on language: 

Line 54: Suggest rephrasing of the statement "and this can be favored by centrosome amplification (29)" 

Perhaps a word like potentiated instead of favored? 

Line 67: Again consider using an alternative to favored "We show that centrosome amplification favors the response to combined Carboplatin and Paclitaxel via multiple mechanisms." 

Favored is in fact used throughout the manuscript text- in my opinion this is not a scientific enough term and would consider replacing with alternative. Line 263: "Centrosome amplification primes for MOMP and sensitizes cells to a diversity of chemotherapies." CA primes to one very specific BCL-XL inhibitor in this section so consider modifying the title of the section. 

Overall, this well-written work extends and provides mechanistic detail to understand the role of CA in priming cells for cytotoxicity in response to commonly used chemo agents in the EOC context. It is a thorough study with sound conclusions drawn from the data provided. It also employs a broad range of assays and techniques to explore the hypotheses from every angle. In view of this, this manuscript is a valuable contribution to the literature on the role of CA in ovarian cancer and its treatment.

Reviewer #3: This is a valuable archival paper that catalogues the effects of combined treatment with paclitaxel and carboplatin predominantly on one ovarian cancer cell line, OVCAR8, in which extra centrosomes can be induced by induced overexpression of Plk4. It systematically examines cellular responses to this drug treatment regimen in control and Plk4 overexpressing cells. Together the experiments show that Plk4-mediated formation of extracentrosomes sensitizes cells to cell death independently of any effect upon spindle multipolarity and chromosome segregation, irregular spindle formation and mitotic errors, and of the DNA damage response. The authors then go on to show that Plk4 over expression results in premature cleavage of Caspase 3 and so favors the apoptotic response. \\This appears to be mediate through increased mitochondrial outer membrane permeabilization. The PIDDosome is believed to contribute to apoptosis in the presence of extra centrosomes through a p%£ mediated pathway. However, in this case, apoptosis appears to be independent of p53 and also of the PIDDosome, as show by deleting a key PIDDosome component. The authors are therefore left with a bit of a mystery in terms of providing a mechanistic explanation of their findings. I do recommend publication of this paper in its present form as the study has been carried out very carefully and it is very important for workers in the field to know what has been tried in attempt to explain the phenomenon of increased cell death following Plk4 overexpression. It does not lead to a new mechanistic discovery but highlights an important phenomenon that we still have to explain.

---

## [Decision Letter · Decision Letter 2]

5 Mar 2024

Dear Renata,

Thank you for your patience while we considered your revised manuscript "Centrosome amplification primes for apoptosis besides promoting multipolar divisions, and potentiates the response to chemotherapy in ovarian cancer" for publication as a Short Report at PLOS Biology. Please accept my apologies for the delays that you have experienced during this round of the peer review process. Your revised study has been evaluated by the PLOS Biology editors, the Academic Editor and by two of the original reviewers at Review Commons.

The reviews are attached below. As you can see, Reviewer’s #1 and #2 agree that the revised manuscript is improved but continue to raise some concerns with the overall strength of the evidence to support the claims. Specifically, Reviewer #1 notes that the revised version does not include experiments to directly visualize tubulin, in order to rule out the occurrence of multipolar spindles upon centrosome amplification. After discussions with the Academic Editor, we agree that more direct evidence as suggested by the reviewer should be included in a revised version. In addition, Reviewer #1 raises concerns with the quality of the images for cytochrome c release in Figure 4 and with the panels in Figure 2 supporting high mis segregation in the presence of carboplatin.

In light of the reviews, we will not be able to accept the current version of the manuscript, but we would welcome re-submission of a much-revised version that takes into account the reviewers' comments. We cannot make any decision about publication until we have seen the revised manuscript and your response to the reviewers' comments.

**IMPORTANT - SUBMITTING YOUR REVISION**

*Re-submission Checklist*

*Published Peer Review*

*PLOS Data Policy*

*Blot and Gel Data Policy*

Best wishes,

Richard

Richard Hodge, PhD

rhodge@plos.org

REVIEWS:

Reviewer #1: The revised manuscript from Edwards et al is improved, though some concerns remain. Some of the previous critiques have been addressed, including acknowledging that it remains unclear how the priming observed in a population in which ~80% of cells have centrosome amplification would operate in patient tumors with ≤3% centrosome amplification. However, multiple concerns raised have been only partially addressed, and some conclusions presented are too strong based on the available evidence. Additional modifications are warranted prior to publication.

1. The last paragraph of the Introduction and the Results on page 7 accurately describe the significant increase in multipolar spindles in PLK4OE cells as contributing to the increase in cell death in response to paclitaxel+carboplatin. However, the summary and discussion continue to emphasize that PLK4 overexpression induces cell death "independently" of mitotic defects and the revised title is confusing. In keeping with their wording in the last paragraph of the Introduction, changing "besides" to "beyond" in the title would increase clarity (ie "Centrosome amplification primes for apoptosis beyond promoting multipolar divisions…). In the abstract, rather than centrosome amplification acting "independently of" (line 34) multipolar divisions it would be more appropriate to indicate this effect is "over and above" or "in addition to" multipolar divisions.

2. The inability to visualize tubulin, as proposed in the revision plan but not accomplished in the revision, remains a significant weakness. Since the primary mitotic phenotype expected from centrosome amplification is (at least transient) multipolar spindles, it is critical to visualize tubulin to rule this out. Fluorescently tagged tubulin, which can be incorporated after transfection or viral infection, is widely used to visualize mitotic spindles and it is unclear why this approach was not used here, where it would be expected to succeed in this cell line. Instead the authors treated cells with fluorescent taxane, which unsurprisingly caused a mitotic phenotype that precluded interpretation of results. The fixed analysis of tubulin after PLK4OE alone (Fig S2) somewhat mitigates this concern, though the finding that the additional centrosomes do not efficiently form spindle poles is surprising given their ability to do so in other contexts, raising questions about the generality of these findings. Conclusions about neglible multipolar divisions and absence of mitotic errors based on timelapse analysis that does not include tubulin should be tempered based on the available evidence.

3. The images for cytochrome C release in Fig 4D and 4G are still not persuasive. The enlargements in 4G are smaller than those in 3D and have more colors, resulting in images that are very difficult to discern. The low contrast between yellow and white is suboptimal. 

4. Line 200: "In both PLK4Ctl and PLK4OE treated with Carboplatin, the main phenotype was an increase in High mis-segregation divisions…(Fig 2A-B and F-G)". These panels don't show the incidence of mitotic phenotypes in the absence of carboplatin and so can't support this conclusion. The example "anaphase" in Fig S2D looks like it is in prometaphase. Though the cell shown has 1 acentric chromosome, it isn't clear how this will lead to high mis-segregation.

5. The revised statement in the first paragraph of the discussion still misleadingly implies that HSET inhibitors have failed in clinical trials when they have not entered clinical trials. It also implies that Aurora A inhibitors have been tested in the clinic in an attempt to induce multipolar divisions when they were tested based on the ability to induce mitotic arrest. This section should be further revised.

Minor comments

6. The chromosome bridge in Fig. 1A is not visible in the current version of the image.

7. The statement on line 194-5, "Carboplatin treated cells have a reduction in proliferation compared to untreated cells (Fig. S1E)", does not appear to refer to Fig. S1E.

8. An image of an "undefined" spindle in Fig S1I appears to be missing.

9. At what time point were the data in Fig 3D-E collected? Fig 3A-B suggests this is likely to be the 48h timepoint, but I do not see this information in the text, figure or legend.

10. The IC50 values for WEHI-539 in PLK4OE and PLK4Ctrl cells appear to be switched in Fig 4A.

11. How can 59% stage II patients be in the cohort comprising both High and Low CNR (Fig S7D, lines 569-570)?

12. S7E and F have exactly the same p value (0.0182), which seems unlikely. Is this correct?

13. The sentence on line 229-231 implies that No missegregation and slight mis-segregation events occur only in the first 24h of carboplatin treatment, and that other types of divisions do not occur during this time period. Is that correct?

14. Line 539 says STING knockout had no influence on sensitivity to WEHI-539 (Fig S6G-H), but Fig S6H indicates it causes a significant decrease in cell death. 

15. On line 38, "occurs" really should be "can occur" because enhanced cell death is in some cases due to chromosome instability.

Reviewer #2: The authors have addressed the points raised in my initial review in Review Commons with the additional expts and addendums to the manuscript :

Reviewer 2, Major comments:

1. The authors state that (Line 133) "the increased multipolarity we observe in presence of the

combined chemotherapy is caused by the effect of Paclitaxel on the capacity of cells to cluster

centrosomes." Could the authors to back up this claim by reanalyzing the imaging data to look for

clustering as a survival mechanism versus inhibition of clustering in Paclitaxel-treated cells? Or

indeed test any of the range of available clustering inhibitors directly on PLK4OE and thus prove

the contribution of clustering to survival?

The experiments completed by the authors address in some way address my concerns about the contribution of centrosome clustering to cell survival in their model system and also show that Paclitaxel increases the incidence of multipolar/undefined mitotic spindles post-CA induction. In the final version, images representing “abnormally looking spindles that we named undefined spindles” should also be included in this figure for clarity. Finally, reword “abnormally” to “abnormal”

2. Fig 3: Results line 229-236 refer to quantification of fragmented nuclei which the authors

interpret as poised for apoptosis. Micronuclei are also quantified- do the authors interpret this

phenotype as advanced apoptosis? There is no mention of apoptotic bodies in the analysis. I

would ask the authors to provide a bank of representative images with explanations to illustrate

their interpretation of the range of morphologies - differences between nuclear fragmentation,

versus micronuclei versus DNA contained in apoptotic bodies.

The authors have adequately addressed my query by highlighting the presence of apoptotic bodies with additional labelling in Fig 3 and providing representative images of the various nuclear morphologies.

3. Fig 6: While the authors have already acknowledged this as a weakness of the study, can the

patient data really be compared to cell line data on CA because inclusion of CNRs between 1.4

and 2 as "high CNR" is questionable given that this ratio represents a completely normal

centrosome complement? Are the authors confident enough in the imaging technique that all

centrosomes are being detected? Can the authors justify the inclusion of the 1.4-2 CNR tumours

by breaking down individual patient data on response to various treatments? Have the authors

tried to analyse the cohort for OS and RFS using only those 9% of tumours exhibiting CA? What

does the analyses of Fig 6 and S6 look like with a CNR cut-off of 2 instead of 1.45? Does the reanalysis

show a better correlation between CNR and FIGO stage?

The authors have satisfactorily addressed all the queries raised in this comment.

4. Although this patient cohort is described in a previous publication, authors should include a

cohort description in a table within supplemental for this manuscript: age range of patients,

number of patients in each stage, size of tumours, and most relevant to this study, treatment

regimens- adjuvant versus neoadjuvant, surgery vs no surgery? How is the cohort selected- sequentially

selected? inclusion/exclusion criteria?

Statement in abstract "we show that high centrosome numbers associate with improved chemo

responses" is too specific as we have no information on the treatment regimens received by the

patients (neo or adjuvant chemo versus surgical/radiological interventions?). Maybe treatment

response would be more appropriate? Were there any cases of Pathological complete (or even

near complete) response in this cohort and if so, what was the CNR in those cases?

The authors have satisfactorily addressed all the queries raised in this comment.

5. The experimental PLK4 overexpression system is an accepted and clean method to induce CA

in vitro. Could the authors comment in the discussion on how they envision CA being induced as

a sensitizing agent in the clinical setting to support the translational aspects of their work?

The authors have satisfactorily answered the query raised in this comment.

Reviewer 2, Minor comments: 

Line 54: Suggest rephrasing of the statement “and this can be favored by centrosome amplification (29)” 

Perhaps a word like potentiated instead of favored?

Line 67: Again consider using an alternative to favored “We show that centrosome amplification favors the response to combined Carboplatin and Paclitaxel via multiple mechanisms.”

Favored is in fact used throughout the manuscript text- in my opinion this is not a scientific enough term and would consider replacing with alternative.

Line 263: “Centrosome amplification primes for MOMP and sensitizes cells to a diversity of chemotherapies.” CA primes to one very specific BCL-XL inhibitor in this section so consider modifying the title of the section.

The authors have satisfactorily addressed all the queries raised in the minor comments section.

---

## [Decision Letter · Decision Letter 3]

5 Jun 2024

Dear Dr Basto,

Thank you for your patience while we considered your revised manuscript "Centrosome amplification primes for apoptosis beyond promoting multipolar divisions, and potentiates the response to chemotherapy in ovarian cancer" for publication as a Short Reports at PLOS Biology. This revised version of your manuscript has been evaluated by the PLOS Biology editors, the Academic Editor and one of the original reviewers.

Based on the reviews, we are likely to accept this manuscript for publication, provided you satisfactorily address the following data and other policy-related requests.

*Please the last remaining comment of Reviewer 2 (see below).

*We would like to suggest a different title to improve readability: 'Centrosome amplification primes ovarian cancer cells for apoptosis and potentiates the response to chemotherapy'.

*Please add the links to the funding agencies in the Financial Disclosure statement in the manuscript details.

*ETHICS STATEMENT:

-- Please include information about the form of consent (written/oral) given for research involving human participants. All research involving human participants must have been approved by the authors' Institutional Review Board (IRB) or an equivalent committee, and must have been conducted according to the principles expressed in the Declaration of Helsinki.

*DATA POLICY:

Regardless of the method selected, please ensure that you provide the individual numerical values that underlie the summary data displayed in the following figure panels as they are essential for readers to assess your analysis and to reproduce it: 1BCDEF, 2ABDEFGHIJK, 3BCEF, 4ACEFHI

*BLOT AND GEL REPORTING REQUIREMENTS:

We require the original, uncropped and minimally adjusted images supporting all blot and gel results reported in an article's figures or Supporting Information files. We appreciate that you've uploaded S1 raw images already. However, upon inspection it seems that the GAPDH controls for S3G and S6F are missing. 

We expect to receive your revised manuscript within two weeks. 

*Published Peer Review History*

*Press*

Sincerely,

Suzanne

Suzanne De Bruijn, PhD, 

Associate Editor

sbruijn@plos.org

PLOS Biology

Reviewer remarks:

Reviewer #2: Once the color scheme in Figure S2I is defined, the revised manuscript will have adequately addressed the points raised in review.

---

## [Editor Report · Decision Letter 4]

17 Jul 2024

Dear Renata,

On behalf of my colleagues and the Academic Editor, Ana Garcia-Saez, I am pleased to say that we can accept your manuscript for publication, provided you address any remaining formatting and reporting issues. These will be detailed in an email you should receive within 2-3 business days from our colleagues in the journal operations team; no action is required from you until then. Please note that we will not be able to formally accept your manuscript and schedule it for publication until you have completed any requested changes.

PRESS

Best wishes, 

Richard

Richard Hodge, PhD 

rhodge@plos.org

PLOS
